# Understanding the Expressivity and Trainability of Fourier Neural Operator: A Mean-Field Perspective

**Takeshi Koshizuka**
Department of Computer Science
The University of Tokyo
koshizuka-takeshi938444@g.ecc.u-tokyo.ac.jp

**Masahiro Fujisawa**
RIKEN AIP
masahiro.fujisawa@riken.jp

**Yusuke Tanaka**
NTT Communication Science Laboratories
ysk.tanaka@ntt.com

**Issei Sato**
Department of Computer Science
The University of Tokyo
sato@g.ecc.u-tokyo.ac.jp

## Abstract

In this paper, we explores the expressivity and trainability of the Fourier Neural Operator (FNO). We establish a mean-field theory for the FNO, analyzing the behavior of the random FNO from an *edge of chaos* perspective. Our investigation into the expressivity of a random FNO involves examining the ordered-chaos phase transition of the network based on the weight distribution. This phase transition demonstrates characteristics unique to the FNO, induced by mode truncation, while also showcasing similarities to those of densely connected networks. Furthermore, we identify a connection between expressivity and trainability: the ordered and chaotic phases correspond to regions of vanishing and exploding gradients, respectively. This finding provides a practical prerequisite for the stable training of the FNO. Our experimental results corroborate our theoretical findings.

## 1 Introduction

The recent surge in interest in solving partial differential equations (PDEs) has led to the use of neural network (NN)-based surrogate models. One promising line of work is the neural operator (NO), which learns the solution operator of PDEs, thereby bypassing the need for mesh dependency. Among the variants of NO, the Fourier neural operator (FNO) (Li et al., 2020c) has gained popularity because of its advantageous cost/accuracy trade-off. The FNO can capture long-distance spatial interactions using the Fourier transform, whereas convolutional neural networks (CNNs) (Wen et al., 2019; Jiang et al., 2021b) and message-passing graph neural networks (GNNs) (Li et al., 2020a,b) are limited to operating solely on local variables. From a computational cost perspective, the Fourier transform is performed in quasi-linear time by the fast Fourier transform (FFT), making it significantly faster than the Transformer (Li et al., 2023).

Despite the widespread use of FNO as an architecture, there is a lack of comprehensive theoretical analysis on its expressivity and trainability. The universal approximation property (Kovachki et al., 2021), recognized as the basic expressivity of the FNO, is well-known; however, the exponential expressivity depending on the weight distribution, which are known for the densely connected network (DCN) (Schoenholz et al., 2016), a.k.a. fully connected network, and CNN (Xiao et al., 2018), remains unexplored. Regarding the trainability of FNO, the training instability in deep FNO has been experimentally reported by Tran et al. (2022), but the causes and conditions of the difficulty have not been clarified either theoretically or experimentally.

38th Conference on Neural Information Processing Systems (NeurIPS 2024).

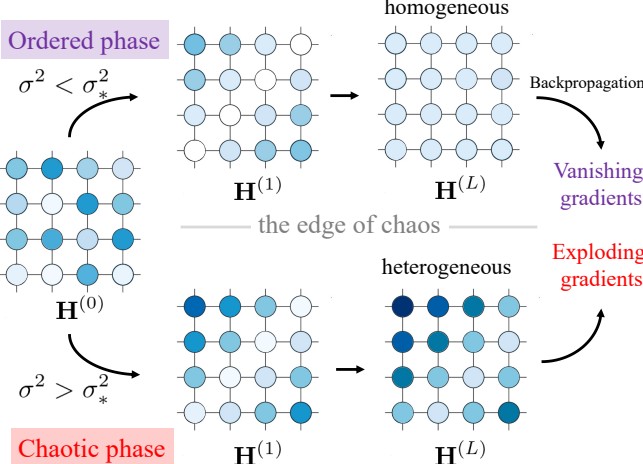

Figure 1: Illustration of ordered-chaos phase transition for the weight initialization parameter $\sigma^2$. In the ordered phase, the spatial hidden representations $\mathbf{H}^{(\ell)}$ on the grid converge to a uniform state during forward propagation and the gradient vanishes during backpropagation. In the chaotic phase, the representations either converge to a distinct state or diverge and the gradient explodes.

We analyze the exponential expressivity (how far can two different input vectors be pulled apart) and trainability (how much gradient explosion on average) of the random FNO from the perspective of whether the network is *ordered* or *chaotic*. This viewpoint is grounded in mean-field theory, an analytical framework for NN established by Poole et al. (2016); Schoenholz et al. (2016); Yang & Schoenholz (2017); Xiao et al. (2018). A network is considered ordered when it brings all representations of two different spatial positions closer together, and chaotic when it drives them apart during forward propagation. Furthermore, a network can only be stably trained when initialized close to the *edge of chaos*, which is the transition point between the ordered phase and the chaotic phase. In fact, He initialization (He et al., 2015) is an example of a commonly used edge of chaos initialization for the DCN with ReLU activation (Burkholz & Dubatovka, 2019).

In this study, we establish a mean-field theory to analyze the expressivity and trainability of the FNO. Our investigation reveals the expressivity of random FNO at initialization by examining the transition point between ordered and chaos phases. The phase transition exhibits FNO-specific characteristics induced by mode truncation, as well as similarities with the characteristics of DCN and CNN. We also find a link between expressivity and trainability: the ordered and chaotic phases correspond to regions of vanishing and exploding gradient, respectively. This discovery offers a practical initialization prerequisite for the stable training of the FNO.

## 2 Background

### 2.1 Fourier Neural Operators

The FNO (Li et al., 2020c) is one of the well-established methods for solving PDEs across many scientific problems (Yang et al., 2021; Wen et al., 2022b; Hwang et al., 2022; Jiang et al., 2021a; Pathak et al., 2022). An $M$-dimensional FNO with the number of hidden features $D$ and the spatial size $N$ for learning the operators between scalar-valued functions is defined as follows.

$$\mathbf{X}^{(\ell+1)} = \phi\left(\mathcal{D}^{(\ell)}\left(\mathbf{X}^{(\ell)}\right) + \mathcal{K}^{(\ell)}\left(\mathbf{X}^{(\ell)}\right)\right), \tag{1}$$

where $\mathbf{X}^{(\ell)} \in \mathbb{R}^{\overbrace{N \times \cdots \times N}^{M} \times D}$ is the $\ell$-th hidden representation, $M$ is the number of spatial dimensions, and $\phi$ is activation. The hidden representations $\mathbf{X}^{(0)}$ and $\mathbf{X}^{(L)}$ are the output of the lifting operator and the input of the projection operator, respectively. The architecture of these operators does not affect our analysis as long as the network stays shallow, as implemented in (Li et al., 2020c). The $\ell$-th densely connected (DC) module $\mathcal{D}^{(\ell)}$ is an affine point-wise map in the physical space and the $\ell$-th

Fourier convolution module $\mathcal{K}^{(\ell)}$ is a parameterized kernel integral operator using Fourier transform. The bias term is considered to be included in either or both modules $\mathcal{K}^{(\ell)}$ and $\mathcal{D}^{(\ell)}$.

Several FNO variants have been developed to address specific challenges, such as geo-FNO (Li et al., 2022) for irregular regions and group equivariant FNO (G-FNO) (Helwig et al., 2023), which maintains equivariance to rotation, reflection, and translation. U-NO (Rahman et al., 2022) and U-FNO (Wen et al., 2022a) integrate FNO with U-Net for multiscale modeling. Additionally, WNO (Tripura & Chakraborty, 2022) utilizes wavelet bases, while CFNO (Brandstetter et al., 2023) enhances the use of geometric relations between different fields and field components through Clifford algebras. Adaptive FNO (Zhao et al., 2022; Guibas et al., 2021) and F-FNO (Tran et al., 2022) have improved computational and memory efficiency through incremental learning and architectural modifications. Other approaches for improving performance include methods with increasing physical inductive bias (Li et al., 2024), data augmentation (Brandstetter et al., 2022), and a variance-preserving weight initialization scheme (Poli et al., 2022).

While numerous new models and learning methods have been proposed, relatively little research has been conducted to understand the intrinsic nature of these methods. Issues such as spectral bias (Zhao et al., 2022) and training instability (Tran et al., 2022) have been reported. Tran et al. (2022) observed that training did not converge even at 24 layers. They successfully addressed the stability and accuracy degradation issues associated with an increase in the number of layers by implementing skip connections behind activation and introducing various training techniques. However, it is still unknown that the theoretical basis for why the original architecture of the FNO has problems with training instability and accuracy degradation.

## 2.2 Mean-field Theory for Neural Networks

The mean-field theory has been used to provide a mathematical framework for understanding the expressivity and trainability of neural networks (Poole et al., 2016; Schoenholz et al., 2016; Yang & Schoenholz, 2017; Hayou et al., 2018; Xiao et al., 2018). A series of papers (Poole et al., 2016; Schoenholz et al., 2016) delved into the average behavior of infinite-width random deep DCN, with weights and biases initialized by a zero-mean Gaussian distribution. The formulation is given below.

$$\boldsymbol{x}^{(\ell)} = \phi\left(\boldsymbol{h}^{(\ell)}\right), \ \boldsymbol{h}^{(\ell)} = \boldsymbol{W}^{(\ell)}\boldsymbol{x}^{(\ell-1)} + \boldsymbol{b}^{(\ell)},$$
$$W_{i,j}^{(\ell)} \overset{i.i.d.}{\sim} \mathcal{N}\left(0, \frac{\sigma^2}{D}\right), \ b_i^{(\ell)} \overset{i.i.d.}{\sim} \mathcal{N}\left(0, \sigma_b^2\right), \tag{2}$$

where $\boldsymbol{x}^{(\ell)} \in \mathbb{R}^D$ is the $\ell$-th hidden representation, $\boldsymbol{W}^{(\ell)} \in \mathbb{R}^{D \times D}$, $\boldsymbol{b}^{(\ell)} \in \mathbb{R}^D$ are the $\ell$-th learnable parameters, and the width is assumed to be sufficiently large $D \gg 1$.

Poole et al. (2016) and Schoenholz et al. (2016) explored the exponential expressivity of random DCN determined by two phases depending on the initial variance parameters $\sigma^2$ and $\sigma_b^2$, as shown in Fig. 5a. Poole et al. (2016) first examined the forward propagation of a random DCN with Tanh activation. They demonstrated that the covariance $\boldsymbol{\Sigma}^{(\ell)}$ of the $\ell$-th pre-activation representations $\boldsymbol{h}^{(\ell)}$ and $\tilde{\boldsymbol{h}}^{(\ell)}$ corresponding to two different inputs $\boldsymbol{x}^{(0)}$ and $\tilde{\boldsymbol{x}}^{(0)}$ are obtained by

$$\forall d \in [D], \ \boldsymbol{\Sigma}^{(\ell)} = \sigma^2 \mathbb{E}\left[\phi\left(h_d^{(\ell-1)}\right) \phi\left(\tilde{h}_d^{(\ell-1)}\right)\right] + \sigma_b^2,$$

where the expectation is taken over the pre-activations $[h_d^{(\ell-1)}, \tilde{h}_d^{(\ell-1)}] \sim \mathcal{N}(\boldsymbol{0}, \boldsymbol{\Sigma}^{(\ell-1)})$. The covariance converges exponentially to a fixed point $\boldsymbol{\Sigma}^*$ determined by parameters $\sigma^2$ and $\sigma_b^2$.

A network is considered *ordered* when it brings two distinct representations closer together, which implies a state of small expressivity. Conversely, a network is *chaotic* when it drives them apart during forward propagation, implying a state of large expressivity. Networks with either excessively small or large expressivity can disrupt the structure of the input: the difference between two distinct inputs quickly becomes indistinguishable in networks with small expressivity, while similarities between inputs are no longer recognized in networks with large expressivity. The network is *ordered* if the initial variance of the weights is small. For larger values, and beyond a certain threshold, the phase shifts, and the network behaves chaotically. This phase shift point is termed *the edge of chaos*.

Subsequently, Schoenholz et al. (2016) discovered the connection between expressivity and trainability of DCN through analysis of the backpropagation behavior. In an ordered network, the expected

value of the gradient norm becomes exponentially small during backpropagation, while it becomes exponentially large in a chaotic network. This implies that the gradient vanishes/explodes in ordered or chaotic networks, respectively. These findings suggest that deep DCN can be stably trained only near the edge of chaos. Schoenholz et al. (2016) also provided an estimate of the maximum depth at which a network can be trained when initialized away from the edge of chaos. These insights are not limited to DCN and similar results have been observed for residual networks (Yang & Schoenholz, 2017) and CNN (Xiao et al., 2018).

# 3 A mean-field theory for FNO

In this section, we establish a mean-field theory of FNO. We demonstrate the exponential expressivity of random FNO by examining the ordered-chaos phase transition during the forward propagation. Furthermore, we identify the connection between expressivity and trainability by concentrating on backward propagation behaviors. Our analysis is an advanced version of the approach developed by Poole et al. (2016); Schoenholz et al. (2016); Yang & Schoenholz (2017); Xiao et al. (2018). In Section 3.1, we outline the problem setup. In Section 3.2, we analyze the forward and backward propagation behavior of random FNO at initialization. In Section 3.3, we discuss the practical prerequisites for initialization to stabilize the training of FNO, leveraging the similarities between FNO and DCN. The proofs for all the lemmas and theorems are provided in Appendices A and B.

## 3.1 Problem setting

Here, we consider a simplified one-dimensional (1D) FNO. Note that our theory is extensively applicable to the original FNO, as discussed in Section 3.3. The simplified 1D FNO, with a depth of $L$, is defined by the number of hidden features $D$, a spatial size $N = 2^m$ (where $m$ is an integer), the number of Fourier modes $K \leq \frac{N}{2} + 1$, two learnable weights $\mathbf{\Theta}^{(\ell,k)} \in \mathbb{R}^{D \times D}$ and $\mathbf{\Xi}^{(\ell,k)} \in \mathbb{R}^{D \times D}$, and a bias $\mathbf{b}^{(\ell)} \in \mathbb{R}^D$. Denote $\phi \colon \mathbb{R} \to \mathbb{R}$ by the non-decreasing activation function. Let $\mathbf{X}^{(\ell)} \in \mathbb{R}^{N \times D}$ and $\mathbf{H}^{(\ell)} \in \mathbb{R}^{N \times D}$ be the post and pre-activation representations defined by

$$\mathbf{X}^{(\ell)} = \phi\left(\mathbf{H}^{(\ell)}\right), \ \mathbf{H}^{(\ell)} = \sum_{k=0}^{K-1} \sqrt{\frac{c_k}{2}} \left(\mathbf{H}^{(\ell,k)} + \overline{\mathbf{H}}^{(\ell,k)}\right) + \mathbf{b}^{(\ell)} \mathbf{1}_N^\top,$$

$$\mathbf{H}^{(\ell,k)} := \mathbf{F}^\dagger \mathbf{D}^{(k)} \mathbf{F} \mathbf{X}^{(\ell-1)} \left(\mathbf{\Theta}^{(\ell,k)} + \sqrt{-1} \mathbf{\Xi}^{(\ell,k)}\right), \quad (3)$$

where $\delta_{a,b}$ is the Kronecker-delta, $c_k = 2 - \delta_{k,0} - \delta_{k,N/2}$ is a constant, $\mathbf{1}_N$ is all-ones column vector with the size $N$, $\overline{\mathbf{H}}^{(\ell,k)}$ is the conjugate of $\mathbf{H}^{(\ell,k)}$ corresponding to the $(N-k)$-th frequency components, $\dagger$ is the transpose conjugate, $\mathbf{F} \in \mathbb{C}^{N \times N}$ is the Discrete Fourier Transform (DFT) matrix defined by $F_{k,n} = \frac{1}{N} \exp(-\frac{2\pi k}{N} n)$, and $\mathbf{D}^{(k)}$ is a diagonal matrix with a 1 at position $D_{k,k}^{(k)}$.

There are two differences from the original FNO proposed by Li et al. (2020a): (1) the DC module is dropped for the simplicity, and (2) $\mathbf{H}^{(\ell,k)}$ is multiplied by $\sqrt{2}$ with respect to $k = 0, \frac{N}{2}$ for appropriate normalization. We assume that the weights of FNO are initialized by i.i.d. samples from Gaussian distribution, *i.e.* $\Theta_{i,j}^{(\ell,k)} \overset{i.i.d.}{\sim} \mathcal{N}(0, \frac{\sigma^2}{2D})$, $\Xi_{i,j}^{(\ell,k)} \overset{i.i.d.}{\sim} \mathcal{N}(0, \frac{\sigma^2}{2D})$, $b_i^{(\ell)} \overset{i.i.d.}{\sim} \mathcal{N}(0, \sigma_b^2)$. For $k = 0, \frac{N}{2}$, the parameter $\mathbf{\Xi}^{(\ell,k)}$ is set to zero exceptionally. For all $d \in [D] = \{0, \ldots, D-1\}$, the pre-activations $\mathbf{H}_{:,d}^{(\ell)} \in \mathbb{R}^N$ are i.i.d. random variables. When $D \gg 1$, by the central limit theorem, the variables $\mathbf{H}_{:,d}^{(\ell)}$ follow Gaussian distribution with mean 0 and covariance matrix $\Sigma_{\alpha,\alpha'}^{(\ell)} := \mathbb{E}_{\Theta^{1:\ell}, \Xi^{1:\ell}} \left[ H_{\alpha,d}^{(\ell)} H_{\alpha',d}^{(\ell)} \right]$, where the expectation is taken over all random variables $[\Theta^{1:\ell}, \Xi^{1:\ell}] := \{\mathbf{\Theta}^{(\ell',k')}, \mathbf{\Xi}^{(\ell',k')}\}_{\ell' \in [\ell], k' \in [K]}$. Our theory can be easily extended to 2D and 3D FNOs.

## 3.2 Expressivity and trainability of FNO

Firstly, the forward propagation of a single input signal with spatial features is described as follows.

**Lemma 3.1** (Iterated map). *For all $d \in [D]$, the covariance $\mathbf{\Sigma}^{(\ell)} := \mathbb{E}_{\Theta^{1:\ell}, \Xi^{1:\ell}} \left[ \mathbf{H}_{:,d}^{(\ell)} \mathbf{H}_{:,d}^{(\ell)}{}^{\top} \right]$ is obtained recursively by the iterated map $\mathcal{C}$ defined by*

$$\Sigma_{\alpha,\alpha'}^{(\ell)} = \underbrace{\sigma^2 \sum_{k=0}^{K-1} c_k \mathbb{E}\left[ \left| [\boldsymbol{F}\phi\left(\mathbf{H}_{:,d}\right)]_k \right|^2 \right] \cos\left( \theta_{\alpha,\alpha'}^{(k)} \right) + \sigma_b^2,}_{=: \mathcal{C}(\mathbf{\Sigma}^{(\ell-1)})_{\alpha,\alpha'}} \tag{4}$$

*where the expectation is taken over the pre-activations $\mathbf{H}_{:,d} \sim \mathcal{N}(0, \mathbf{\Sigma}^{(\ell-1)})$, $\theta_{\alpha,\alpha'}^{(k)} := \frac{2\pi k}{N}(\alpha - \alpha')$ represents the scaled positional difference.*

The indices $\alpha$ and $\alpha'$ correspond to different spatial locations as with the mean-field theory for CNN (Xiao et al., 2018). Note that $[\boldsymbol{F}\phi\left(\mathbf{H}_{:,d}\right)]_k$ is the $k$-th Fourier modes of the post-activation representation. When applying DCN (Poole et al., 2016; Schoenholz et al., 2016) or CNN (Xiao et al., 2018) to the spatial signal, the iterated map depends only on local spatial locations, while in the case of FNO, the iterated map depends on all spatial locations because of the global Fourier convolution. In addition, only periodic spatial correlations with shift-invariant are propagated, and high-frequency components exceeding mode $K$ are eliminated.

Next, we explore the fixed point $\mathbf{\Sigma}^*$ of the iterated map $\mathcal{C}$ satisfying $\mathbf{\Sigma}^* = \mathcal{C}(\mathbf{\Sigma}^*)$. By linearizing the dynamics of signal propagation around this fixed point and analyzing the stability and rate of convergence to the fixed point, we can determine the depth to which each component of the input can propagate. Schoenholz et al. (2016) showed that the iterated map of DCN defined in Eq. (2) has a fixed point of the form:

$$\mathbf{\Sigma}^* = q^* \boldsymbol{I}_N + q^* c^* (\mathbf{1}_N \mathbf{1}_N^{\top} - \boldsymbol{I}_N), \tag{5}$$

where $q^*, c^*$ are the fixed points of variance and correlation, and $\boldsymbol{I}_N$ is the identity matrix. Meanwhile, Xiao et al. (2018) showed that any fixed point for the iterated map of the DCN is also a fixed point for that of CNN. We show that random FNO has the same fixed points of the form of Eq. (5) with $c^* = 1$ in the following lemma.

**Lemma 3.2** (Exsistance of fixed points). *When a random DCN defined by Eq. (2) has the fixed point $(q^*, c^* = 1)$ for the initial parameters $(\sigma^2, \sigma_b^2)$, then a random simplified FNO defined by Eq. (3) has a fixed point $\mathbf{\Sigma}^*$ of the form*

$$\mathbf{\Sigma}^* = q^* \boldsymbol{I}_N + q^* c^* (\mathbf{1}_N \mathbf{1}_N^{\top} - \boldsymbol{I}_N) = q^* \mathbf{1}_N \mathbf{1}_N^{\top}.$$

Lemma 3.2 indicates that the fixed point for the iterated map $\mathbf{\Sigma}^*$ of the DCN serves as a fixed point for the iterated map of the simplified FNO (as well as CNN). To analyze the stability and convergence rate, we linearly approximate the C-map around the fixed point $\mathbf{\Sigma}^*$, i.e., $\mathcal{C}(\mathbf{\Sigma}) \approx \mathbf{\Sigma}^* + J_{\mathbf{\Sigma}^*}(\mathbf{\Sigma} - \mathbf{\Sigma}^*)$, where $J_{\mathbf{\Sigma}^*}$ is the Jacobian linear map of the iterated map defined in Eq. (15). We then derive the eigenvalues and eigenvectors for the Jacobian linear map $J_{\mathbf{\Sigma}^*}$ as follows.

**Definition 3.3.**

$$\chi_{q^*} := \sigma^2 \mathbb{E}\left[ \phi'^2(H_{\alpha,d}) + \phi''(H_{\alpha,d})\phi(H_{\alpha,d}) \right], \tag{6}$$

$$\chi_{c^*} := \sigma^2 \mathbb{E}[\phi'(H_{\alpha,d})\phi'(H_{\alpha',d})], \tag{7}$$

$$\chi_{\kappa} := \frac{\sigma^2}{2} \mathbb{E}\left[ \phi''(H_{\alpha,d})\phi(H_{\alpha',d}) + \phi(H_{\alpha,d})\phi''(H_{\alpha',d}) \right] + \sigma^2 \mathbb{E}\left[ c^* \phi'(H_{\alpha,d})\phi'(H_{\alpha',d}) \right], \tag{8}$$

where the expectation is taken over the pre-activations $\mathbf{H}_{:,d} \sim \mathcal{N}(0, \mathbf{\Sigma}^*)$, and $\phi'$, $\phi''$ are the first- and second-order derivatives of the activation $\phi$.

The bases $\boldsymbol{\psi}$, $\boldsymbol{\psi}^{(1)}$, ..., $\boldsymbol{\psi}^{(K-1)} \in \mathbb{R}^{N \times N}$ using the above quantities are defined below.

$$\psi_{\beta,\beta'} := 1 - \frac{1}{N}\left( \frac{\chi_{\kappa} + \chi_{c^*} - \chi_{q^*}}{\chi_{\kappa}} \right) \sum_{s=0}^{K-1} c_s \cos\left( \theta_{\beta,\beta'}^{(s)} \right),$$

$$\psi_{\beta,\beta'}^{(k)} := \cos\left( \theta_{\beta,\beta'}^{(k)} \right) - \frac{1}{\sum_{s=0}^{K-1} c_s} \sum_{s=0}^{K-1} c_s \cos\left( \theta_{\beta,\beta'}^{(s)} \right). \tag{9}$$

From Lemma A.4, $K-1$ matrices in $\{\boldsymbol{\psi}^{(k)}\}_{k\in[K]\setminus\{0\}}$ are eigenbases with the eigenvalue $\chi_{c^*}$ of the Jacobian linear map. From Lemma A.5, the matrix $\boldsymbol{\psi}$ is the eigenbases with the eigenvalue $\chi$ of the Jacobian linear map. Since the rank of the Jacobian linear map is at most K (Lemma A.3), the deviation from the fixed point $\boldsymbol{\Sigma}^{(\ell)}-\boldsymbol{\Sigma}^*$ is spanned by K-dimensional eigenspace $\mathrm{span}\left(\{\boldsymbol{\psi}^{(k)}\}_{k\in[K]\setminus\{0\}}\cup\{\boldsymbol{\psi}\}\right)$. Then, the fixed point stability and the convergence rate are shown in the following theorem.

**Theorem 3.4** (Exponential expressivity). *Let $\boldsymbol{E}^{(\ell)}:=\boldsymbol{\Sigma}^{(\ell)}-\boldsymbol{\Sigma}^*$ be the deviation from the fixed point at the $\ell$-th layer. Suppose that the deviation at the first layer is decomposed as $\boldsymbol{E}^{(0)}=\epsilon\boldsymbol{\psi}+\sum_{k=1}^{K-1}\epsilon_k\boldsymbol{\psi}^{(k)}+\boldsymbol{e}$. The scalars $\epsilon$, $\epsilon_k$ represent the scale of the perturbation for each eigencomponent of the linearly approximated map $\boldsymbol{E}^{(\ell)}\mapsto\boldsymbol{E}^{(\ell+1)}$. The component $\boldsymbol{e}\in\mathbb{R}^{N\times N}$ belongs to the orthogonal complements of the space $\mathrm{span}\left(\{\boldsymbol{\psi},\boldsymbol{\psi}^{(1)},\ldots,\boldsymbol{\psi}^{(K-1)}\}\right)$.*

*Then, the deviation at the $\ell$-th layer is obtained by*

$$\boldsymbol{E}^{(\ell)}=\chi^\ell\epsilon\boldsymbol{\psi}+\sum_{k=1}^{K-1}\chi_{c^*}^\ell\epsilon_k\boldsymbol{\psi}^{(k)},\tag{10}$$

$$\chi:=\frac{1}{N}\sum_{s=0}^{K-1}c_s\chi_{q^*}+\left(1-\frac{1}{N}\sum_{s=0}^{K-1}c_s\right)(\chi_\kappa+\chi_{c^*}).$$

*In particular, when the Fourier mode is $K=\frac{N}{2}+1$, Eqs. (9) and (10) reduce to the following.*

$$\boldsymbol{E}^{(\ell)}=\chi_{q^*}^\ell\epsilon\boldsymbol{\psi}+\sum_{k=1}^{K-1}\chi_{c^*}^\ell\epsilon_k\boldsymbol{\psi}^{(k)},\tag{11}$$

$$\forall\beta,\beta'\in[N],\ \psi_{\beta,\beta'}=1,\ \psi_{\beta,\beta'}^{(k)}=\cos\left(\theta_{\beta,\beta'}^{(k)}\right)-\delta_{\beta,\beta'}.$$

Theorem 3.4 shows the expressivity of the FNO, which is characterized by the ordered-chaos phases and varies exponentially with respect to the number of layers. Theorem 3.4 indicates that the asymptotic behavior of the zero-frequency deviation is mostly determined by $\chi$ and the periodic deviation is determined by $\chi_{c^*}$. If $\chi<1$ and $\chi_{c^*}<1$, the fixed point is stable as the deviation from the fixed point converges exponentially to zero. When the fixed point remains stable at $c^*=1$, a random network exists in an ordered phase, where all spatial representations are correlated in an asymptotic manner. Conversely, when the fixed point with $c^*=1$ becomes unstable, the network transitions into a chaotic phase, exhibiting behavior dependent on the activation function $\phi$. The boundary between these two phases is referred to as *the edge of chaos*.

The convergence rates $\chi_{q^*}$ and $\chi_{c^*}$ are the same as the convergence rates of the variance and correlation to the fixed point for DCN (Schoenholz et al., 2016) and CNN (Xiao et al., 2018). However, only periodic spatial correlations are propagated in the FNO, resulting in a different eigenspace of the map $\boldsymbol{E}^{(\ell)}\mapsto\boldsymbol{E}^{(\ell+1)}$ compared to the DCN and CNN. In DCN, the deviation belongs to a vector space with dimension $\frac{N(N-1)}{2}$ in DCN, whereas in FNO, the dimension is $K$, or at most $\frac{N}{2}+1$. CNN possess diagonal eigenspaces associated with eigenvalues $\chi_{q^*}$ and non-diagonal eigenspaces associated with eigenvalues $\chi_{c^*}$. In contrast, FNOs without mode truncation exhibit a similarity, possessing eigenspaces $\chi_{q^*}$ for zero-frequency and eigenspaces $\chi_{c^*}$ for k-frequencies with diagonal components removed. Furthermore, mode truncation increases the convergence rate of zero-frequency deviation from $\chi_{q^*}$ to $\chi$ and affects all eigenbases as well. For further discussions on the similarities between CNN and FNO, please refer to Appendix C. A visualization of the covariance of the FNO with Tanh and ReLU activations is shown in Appendix F.

Finally, we demonstrate the connection between expressivity and trainability. By examining the covariance of the gradient in each layer during backpropagation, we investigate the conditions under which training is stable without gradient vanishing or exploding.

**Theorem 3.5** (Trainability). *Let $\tilde{\boldsymbol{\Sigma}}^{(\ell)}\in\mathbb{R}^{N\times N}$ be the gradient covariance with respect to some loss $\mathcal{L}$, e.g. mean squared error, at the $\ell$-th layer. Suppose that the gradient covariance at the L-th layer is decomposed as $\tilde{\Sigma}_{\alpha,\alpha'}^{(L)}=\sum_{k=0}^{K-1}\tilde{\epsilon}_k\cos\left(\theta_{\alpha,\alpha'}^{(k)}\right)+\tilde{\boldsymbol{e}}$, where $\tilde{\epsilon}_k$ is the coefficient of each basis and $\tilde{\boldsymbol{e}}$*

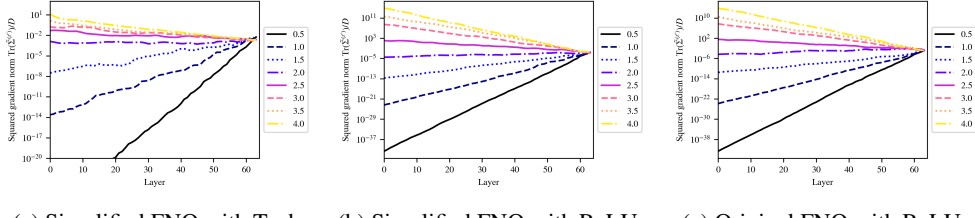

(a) Simplified FNO with Tanh    (b) Simplified FNO with ReLU    (c) Original FNO with ReLU

Figure 2: Average gradient norm $\text{Tr}(\tilde{\boldsymbol{\Sigma}}^{(\ell)})/D$ during the backpropagation of several FNOs plotted as a function of layer $\ell$. Each line corresponds to the result of different initial values of $\sigma^2$ from $0.5$ to $4.0$ in increments of $0.5$. The x-axis is the layer and the y-axis is the log-scale of the gradient norm. Depending on the value of $\sigma^2$, the gradient norm increases or decreases consistently as the gradient propagates to shallower layers.

*belongs to the orthogonal complements of* $\text{span}(\{\cos\left(\theta_{\alpha,\alpha'}^{(k)}\right)\}_{k=0}^{K-1})$. *Then, the gradient covariance at the $\ell$-th layer is obtained by*

$$\tilde{\Sigma}_{\alpha,\alpha'}^{(\ell)} = \sum_{k=0}^{K-1} \chi_{c^*}^{L-\ell} \tilde{\epsilon}_k \cos\left(\theta_{\alpha,\alpha'}^{(k)}\right).$$

Theorem 3.5 shows that gradient vanishing occurs when $\chi_{c^*} < 1$ (ordered phase) and gradient explosion occurs when $\chi_{c^*} > 1$ (chaos phase). Thus, stable training of the FNO can be achieved close to *the edge of chaos* by setting the initial parameter $\sigma^2$ to satisfy $\chi_{c^*} \approx 1$. We specifically present the initial parameter choices that achieve *the edge of chaos* for several FNOs in Section 3.3.

When $c^* = 1$, there is no change in the dynamics during backpropagation due to mode truncation. When using the full mode $K = \frac{N}{2} + 1$, the condition $\chi_{c^*} = 1$ always achieves *the edge of chaos*, which is consistent with the results for the DCN and CNN. Despite the architectural and iterative map differences among FNO, DCN, and CNN, Theorem 3.4 and Theorem 3.5 demonstrate the similarities in the random behavior of FNO, DCN, and CNN. This allows existing results based on mean-field theory to be applied to the FNO.

### 3.3    Initialization requirements for stable training

For stable training, Theorem 3.5 suggests the necessity of initializing FNO near *the edge of chaos*, *i.e.*, initializing FNO so that $\chi_{c^*} \approx 1$. In this section, we present the initial parameter choices that achieve *the edge of chaos* for several FNOs, each with slightly different architectures such as activation functions. Furthermore, the behavior of the gradient norm $\text{Tr}(\tilde{\boldsymbol{\Sigma}}^{(\ell)})/D$ as a function of layer $\ell$ are visualized in Fig. 2 for several variants of random FNO with different initialization parameters $\sigma^2$. We used FNO with a width of $D = 32$ and a number of layers $L = 64$, and for simplicity, we computed the absolute value of the output as the loss with respect to the input sampled from the standard normal distribution.

**Simplified FNO with Tanh activation**.
The behavior of $\chi_{c^*}$ for the parameters $\sigma^2$ and $\sigma_b^2$ of the Tanh network has been extensively studied by Poole et al. (2016); Schoenholz et al. (2016). The phase diagram drawn by Pennington et al. (2017) is shown in Fig. 5a. By using parameters $(\sigma^2, \sigma_b^2)$ around the two phase boundaries of ordered and chaotic that achieve $\chi_{c^*} = 1$, the training of the simplified FNO with Tanh activation can be stabilized. Figure 2a depicts the behavior of the gradient backpropagation in the simplified FNO with Tanh activation and the bias parameter being $\sigma_b^2 = 0.1$. Figure 2a shows that when $\sigma^2 \lesssim 2$, the gradient diminishes exponentially; otherwise, it explodes exponentially.

**Simplified FNO with ReLU activation**.
The iterated map $\mathcal{C}$ of the DCN with ReLU activation is given by Cho & Saul (2009) as follows.

$$q^{(\ell+1)} = \frac{1}{2}\sigma^2 q^{(\ell)} + \sigma_b^2, \tag{12}$$

$$c^{(\ell+1)}q^{(\ell+1)} = \frac{1}{2}\sigma^2 q^{(\ell)}\mathbb{J}_1\left(\frac{c^{(\ell)}q^{(\ell)}}{q^{(\ell)}}\right) + \sigma_b^2, \tag{13}$$

$$\mathbb{J}_1(c) = \frac{1}{\pi}\left(\sqrt{1-c^2} + (\pi - \arccos(c))c\right),$$

The edge of chaos initialization for the DCN with ReLU activation is known as He initialization (He et al., 2015), which sets the initial variance parameter as $\sigma^2 = 2$ and initial bias as $b_i^{(\ell)} = 0$ for Eq. (2). From the similarity of the DCN and the FNO, we can derive the FNO version of the He initialization that achieves $\chi_{c^*} = 1$ by setting $\sigma^2 = 2$, $b_i^{(\ell)} = 0$ for Eq. (3). The He initialization scheme for the simplified FNO with the activation $\phi = \text{ReLU}$ is derived as follows.

$$\Theta_{i,j}^{(\ell,k)} \overset{i.i.d.}{\sim} \mathcal{N}(0, D^{-1}), \ \Xi_{i,j}^{(\ell,k)} \overset{i.i.d.}{\sim} \mathcal{N}(0, D^{-1}).$$

Figure 2b demonstrates that the choice of $\sigma^2 = 2$ preserves the magnitude of the gradient norm during backpropagation of deep simplified FNO with ReLU activation.

**Original FNO**.
In the original architecture of the FNO proposed by Li et al. (2020c), the DC module is used together with the Fourier convolution module as shown in Eq. (1). We initialize the weights of both layers consistently as follows. For all $\ell \in [L]$, $k \in [K]$, and $i, j \in [D]$,

$$\Theta_{i,j}^{(\ell,k)} \overset{i.i.d.}{\sim} \mathcal{N}\left(0, \frac{\sigma^2}{4D}\right), \ \Xi_{i,j}^{(\ell,k)} \overset{i.i.d.}{\sim} \mathcal{N}\left(0, \frac{\sigma^2}{4D}\right),$$

$$W_{i,j}^{(\ell)} \overset{i.i.d.}{\sim} \mathcal{N}\left(0, \frac{\sigma^2}{2D}\right), \ b_i^{(\ell)} \sim \mathcal{N}\left(0, \sigma_b^2\right).$$

From the similarity of the initial network behavior of the FNO and the DCN, the fixed point with $c^* = 1$ of the simplified FNO is also a fixed point of the original FNO. In the neighborhood of the fixed point $\Sigma^*$, the eigencomponents spanned by $\{\psi^{(k)}\}_{k=1}^K$ will decay or increase at the rate of $\chi_{c^*}$. Following the derivation of Theorem 3.5, the eigenvalues of the cos function eigencomponents of the gradient covariance are also $\chi_{c^*}$. These results show that the edge of chaos initialization scheme can be used for the original FNO with each activation function. Figure 2c shows that the original FNO with ReLU activation and the parameter fixed as $b_i^{(\ell)} = 0$ exhibits similar backpropagation behavior as the simplified FNO, *i.e.*, $\sigma^2 \approx 2$ is an appropriate choice.

## 4 Experiments

In this section, we experimentally demonstrate that deep FNO training requires appropriate initialization settings on a variety of datasets, consistent with the theory discussed in Section 3.

### 4.1 Datasets

We evaluated three models on commonly used PDEs: the advection equation, Burgers' equation, Darcy Flow equation, and incompressible Navier-Stokes (NS) equation. All datasets were generated by numerical simulations used in (Takamoto et al., 2022; Li et al., 2020c) and are publicly available. A summary of the dataset is provided in Table 1, and more details are given in Appendix D.

**Advection equation and Burgers' equation**. The linear advection equation for the function $u(x,t)$ is given by

$$\partial_t u(x,t) + \beta \partial_x (u(x,t)/2) = 0, \ u(x,0) = u_0(x),$$

where $u_0$ is the initial condition and $\beta = 2.0$ is an advection speed. The non-linear Burgers' equation for the function $u(x,t)$ is given by

$$\partial_t u(x,t) + \partial_x \left(u^2(x,t)/2\right) = \nu \partial_{xx} u(x,t), \ u(x,0) = u_0(x),$$

Table 1: Overview of dataset with number of spatial dimensions $M$, time dependence, spatial resolution $N_s = N \times \cdots \times N$, temporal resolution $N_t$, and number of samples for training, validation, and testing.

| PDE | $M$ | time | $N_s$ | $N_t$ | Number of samples (Train / Val / Test) |
|---|---|---|---|---|---|
| advection | 1 | - | 64 | - | 8000 / 1000 / 1000 |
| Burgers' | 1 | - | 64 | - | 8000 / 1000 / 1000 |
| Darcy flow | 2 | - | $64 \times 64$ | - | 900 / 100 / 100 |
| Navier-Stokes ($\nu = 1e\text{-}3$) | 2 | ✓ | $64 \times 64$ | 25 | 1000 / 100 / 100 |
| Navier-Stokes ($\nu = 1e\text{-}4$) | 2 | ✓ | $64 \times 64$ | 20 | 8000 / 1000 / 1000 |
| Navier-Stokes ($\nu = 1e\text{-}5$) | 2 | ✓ | $64 \times 64$ | 20 | 1000 / 100 / 100 |

where $u_0$ is the initial condition and $\nu = 4.0$ is the diffusion coefficient.

**Darcy Flow equation**. The Darcy Flow equation for the function $u(x)$ with a Dirichlet boundary is given by

$$-\nabla \cdot (a(x)\nabla u(x)) = f(x) \quad (x \in (0,1)^2), \; u(x) = 0 \quad (x \in \partial(0,1)^2),$$

where $a$ is the diffusion coefficient and $f(x) = 1$ is the forcing function.

**Incompressible Navier-Stokes equation**. The 2D NS equation on the unit torus is defined by

$$\partial_t \omega(x,t) + u(x,t) \cdot \nabla \omega(x,t) = \nu \nabla^2 \omega(x,t) + f(x), \; \nabla \cdot u(x,t) = 0, \; \omega(x,0) = \omega_0(x),$$

where $\omega(x,t)$ is the vorticity, $\omega_0$ is the initial vorticity, $u(x,t)$ is the velocity field for any $r > 0$, $\nu \in \mathbb{R}_+$ is the viscosity, and $f$ is the external forcing function defined by $f(x) = 0.1\left(\sin(2\pi(x_1 + x_2)) + \cos(2\pi(x_1 + x_2))\right)$. We experimented with the viscosities $\nu = 1e\text{-}3, 1e\text{-}4, 1e\text{-}5$. For the data with $\nu = 1e\text{-}4$, the time resolution was also downsampled by half.

### 4.2 Experimental Settings

In the experiments on the 1D advection and Burgers' equation, we compared the results of simplified FNOs defined in Eq. (3) with Tanh and ReLU activations for varying number of layer $L$ and initial parameters $\sigma^2$. In the experiments on the 2D Darcy Flow and NS equations, we compared the results of the original FNO (Li et al., 2020c) as shown in Eq. (1) for varying number of layer $L$ and initial parameters $\sigma^2$. All models have a width of $D = 32$ and the Fourier mode of $K = 12$. When using Tanh as the activation, we fixed the initial bias parameter $\sigma_b^2 = 0.1$, and when using ReLU activation, we fixed the initial bias $b_i^{(\ell)} = 0$. We used the AdamW optimizer and cosine annealing scheduler. Training was stopped early at the epoch of minimal normalized mean squared error (nMSE) on the validation data. Details of the experimental setup are given in Appendix D. In the experiments on the NS equation, we trained the original FNO with the autoregressive scheme using a teacher-forcing technique, input normalization, and regularization that adds small-Gaussian noise to the input (Tran et al., 2022). During the evaluation phase, only the initial 10 steps are provided as input, and the rollout results of all subsequent steps are evaluated. For all tasks, the mean squared error (MSE) is used as the training loss and nMSE as the validation and testing (Li et al., 2020a; Tran et al., 2022).

### 4.3 Results

Heatmaps of the training loss measured by MSE at the last epoch and the test performance measured by nMSE on six different datasets, for different depths $L$ and initial parameters $\sigma^2$, are shown in Figs. 3 and 4, respectively. Despite the differences in architectures and datasets, Figure 3 shows the same trend supporting the theory in all experiments. As the number of layers $L$ increases, the range of acceptable initial $\sigma^2$ value settings becomes narrower, and initialization near *the edge of chaos* ($\sigma^2 \approx 2$) is required for stable training of deep FNO. Detailed analyses on training loss and test performance are presented in Appendix E.1 and Appendix E.2, respectively.

## 5 Conclusion

In this study, we developed a mean-field theory for FNO. We showed the expressivity and trainability of the FNO, which is characterized by the ordered-chaos phases. Furthermore, we observed both

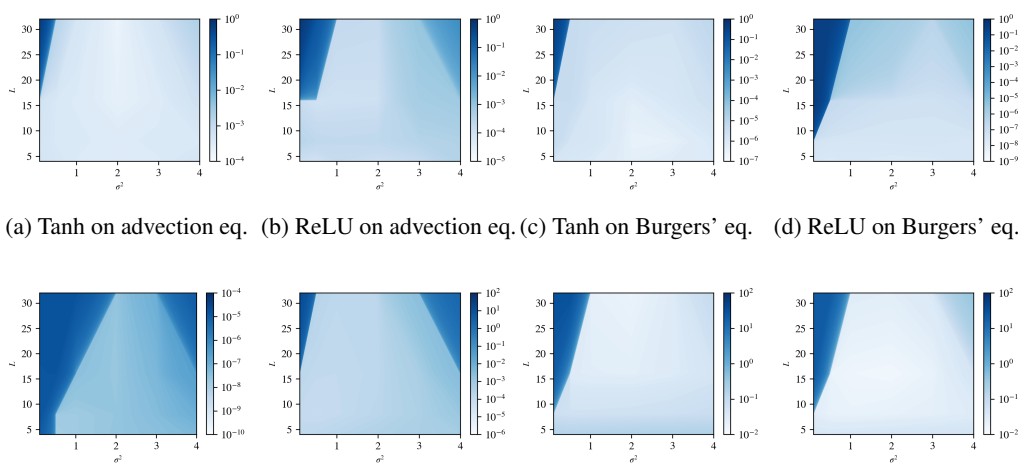

(a) Tanh on advection eq. (b) ReLU on advection eq. (c) Tanh on Burgers' eq. (d) ReLU on Burgers' eq.

(e) ReLU on Darcy Flow (f) ReLU on NS eq. (1e-3) (g) ReLU on NS eq. (1e-4) (h) ReLU on NS eq. (1e-5)

Figure 3: Training loss of FNOs at last epoch for four distinct PDEs. **(a, b):** the advection equation, **(c, d):** the Burgers' equation, **(e):** Darcy Flow, **(f-h):** the NS equation. The heatmaps represents the training loss values for varying depth $L \in \{4, 8, 16, 32\}$ and initial weight parameter $\sigma^2 \in \{0.1, 0.5, 1.0, 2.0, 3.0, 4.0\}$, with lighter colors signifying lower training loss. The presented results are the mean training loss at the last epoch over three different seeds.

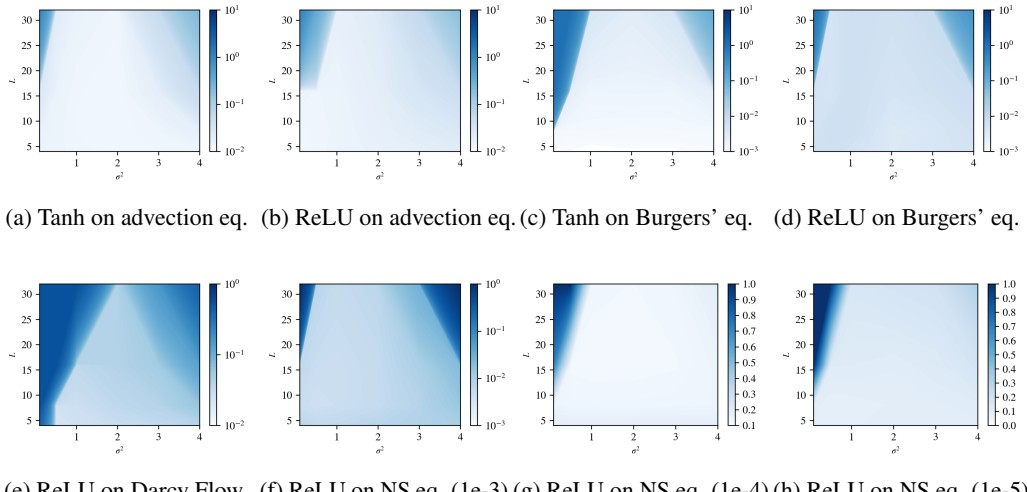

(a) Tanh on advection eq. (b) ReLU on advection eq. (c) Tanh on Burgers' eq. (d) ReLU on Burgers' eq.

(e) ReLU on Darcy Flow (f) ReLU on NS eq. (1e-3) (g) ReLU on NS eq. (1e-4) (h) ReLU on NS eq. (1e-5)

Figure 4: nMSE of FNOs on test datasets for four distinct PDEs. **(a, b):** the advection equation **(c, d):** the Burgers' equation, **(e):** Darcy Flow, **(f-h):** the NS equation. The heatmaps for each nMSE correspond to the results of each heatmap of training loss in Fig. 3. The lighter colors, the better the test performance. The presented results are the mean nMSE calculated over three different seeds.

unique FNO-specific behaviors caused by mode truncation, as well as common behaviors akin to those of DCN. With our analysis as a basis, we identified the necessity of initializing FNO near *the edge of chaos* for stable training of the FNO. Experimental results supported our theoretical results.

A limitation of our analysis is that it is limited to the network at initialization and does not address the stability of the entire optimization process. While we do not provide sufficient conditions for stable training, we do offer one necessary condition for achieving stable training. Future work may consider a mean-field analysis of the FNO when using skip-connection (Tran et al., 2022), Dropout and batch normalization, as well as initialization methods that ensure the input-output Jacobian of the FNO satisfies dynamical isometry (Pennington et al., 2017, 2018).

## Acknowledgement

This work was supported by JSPS KAKENHI Grant Number 24H00709 Japan, JST ASPIRE Grant Number JPMJAP2329, RIKEN Special Postdoctoral Researcher Program, JST ACT-X Grant Number JPMJAX210K, and JST ACT-X Grant Number JPMJAX210D.

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

Figure 5: Ordered-chaos phase transition diagram for the DCN

## A    Proof of Theorem 3.4

**Theorem 3.4** (Exponential expressivity). *Let $\boldsymbol{E}^{(\ell)} := \boldsymbol{\Sigma}^{(\ell)} - \boldsymbol{\Sigma}^*$ be the deviation from the fixed point at the $\ell$-th layer. Suppose that the deviation at the first layer is decomposed as $\boldsymbol{E}^{(0)} = \epsilon\boldsymbol{\psi} + \sum_{k=1}^{K-1} \epsilon_k \boldsymbol{\psi}^{(k)} + \boldsymbol{e}$. The scalars $\epsilon$, $\epsilon_k$ represent the scale of the perturbation for each eigencomponent of the linearly approximated map $\boldsymbol{E}^{(\ell)} \mapsto \boldsymbol{E}^{(\ell+1)}$. The component $\boldsymbol{e} \in \mathbb{R}^{N \times N}$ belongs to the orthogonal complements of the space $\mathrm{span}\left(\{\boldsymbol{\psi}, \boldsymbol{\psi}^{(1)}, \ldots, \boldsymbol{\psi}^{(K-1)}\}\right)$.*

*Then, the deviation at the $\ell$-th layer is obtained by*

$$\boldsymbol{E}^{(\ell)} = \chi^\ell \epsilon\boldsymbol{\psi} + \sum_{k=1}^{K-1} \chi_{c^*}^\ell \epsilon_k \boldsymbol{\psi}^{(k)}, \tag{10}$$

$$\chi := \frac{1}{N} \sum_{s=0}^{K-1} c_s \chi_{q^*} + \left(1 - \frac{1}{N} \sum_{s=0}^{K-1} c_s\right) (\chi_\kappa + \chi_{c^*}).$$

*In particular, when the Fourier mode is $K = \frac{N}{2} + 1$, Eqs. (9) and (10) reduce to the following.*

$$\boldsymbol{E}^{(\ell)} = \chi_{q^*}^\ell \epsilon\boldsymbol{\psi} + \sum_{k=1}^{K-1} \chi_{c^*}^\ell \epsilon_k \boldsymbol{\psi}^{(k)}, \tag{11}$$

$$\forall \beta, \beta' \in [N],\ \psi_{\beta,\beta'} = 1,\ \psi_{\beta,\beta'}^{(k)} = \cos\left(\theta_{\beta,\beta'}^{(k)}\right) - \delta_{\beta,\beta'}.$$

*Proof.* The theorem is obtained by the eigenvalue analysis on the the first-order Taylor approximation of the iterated map $\mathcal{C}$ at the fixed point $\boldsymbol{\Sigma}^*$. The Jacobian matrix $J(\boldsymbol{\Sigma}^*) \in \mathbb{R}^{N^2 \times N^2}$ of the iterated map at the fixed point and the Jacobian linear map $J_{\boldsymbol{\Sigma}^*}(\cdot): \mathbb{R}^{N \times N} \to \mathbb{R}^{N \times N}$ are defined as follows.

$$[J(\boldsymbol{\Sigma}^*)]_{(\alpha,\alpha'),(\beta,\beta')} := \left.\frac{\partial[\mathcal{C}(\boldsymbol{\Sigma})]_{\alpha,\alpha'}}{\partial \Sigma_{\beta,\beta'}}\right|_{\boldsymbol{\Sigma}=\boldsymbol{\Sigma}^*}, \tag{14}$$

$$\forall \boldsymbol{\Sigma} \in \mathbb{R}^{N \times N},\ J_{\boldsymbol{\Sigma}^*}(\boldsymbol{\Sigma}) := \mathrm{mat}\left(J(\boldsymbol{\Sigma}^*) \mathrm{vec}(\boldsymbol{\Sigma})\right), \tag{15}$$

where $\mathrm{vec}$ performs the vectorization, *i.e.* transforming an $N \times N$ matrix into a vector of size $N^2$, and $\mathrm{mat}$ performs the inverse operation of $\mathrm{vec}$.

From Lemma A.3, $K - 1$ matrices in $\{\mathrm{vec}(\boldsymbol{\psi}^{(k)})\}_{k \in [K] \setminus \{0\}}$ are eigenbases with the eigenvalue $\chi_{c^*}$ of the Jacobian linear map. From Lemma A.4, the matrix $\boldsymbol{\psi}$ is the eigenbases with the eigenvalue $\chi$ of the Jacobian linear map. Since the sets of $K$ matrices in $\{\boldsymbol{\psi}^{(k)}\}_{k \in [K] \setminus \{0\}} \cup \{\boldsymbol{\psi}\}$ are linearly

independent (yet non-orthogonal) in $\mathbb{R}^{N \times N}$, the subspace $\mathrm{span}\left(\{\boldsymbol{\psi}^{(k)}\}_{k \in [K] \backslash \{0\}} \cup \{\boldsymbol{\psi}\}\right)$ is the $K$-dimensional eigenspace of the Jacobian linear map. From Lemma A.2, the rank of the Jacobian matrix $J(\boldsymbol{\Sigma}^*)$ at the fixed point $\boldsymbol{\Sigma}^*$ is at most $K$, thereby the rank of the Jacobian linear map in Eq. (15) is at most $K$. Therefore, we have

$$\forall \boldsymbol{e} \in \mathrm{span}\left(\{\boldsymbol{\psi}^{(k)}\}_{k \in [K] \backslash \{0\}} \cup \{\boldsymbol{\psi}\}\right)^{\perp}, \quad J_{\boldsymbol{\Sigma}^*}(\boldsymbol{e}) = \boldsymbol{O}.$$

$$\mathcal{C}(\boldsymbol{\Sigma}^* + \boldsymbol{E}^{(\ell-1)}) \approx \boldsymbol{\Sigma}^* + J_{\boldsymbol{\Sigma}^*}(\boldsymbol{E}^{(\ell-1)})$$

$$= \boldsymbol{\Sigma}^* + \left(\chi^{\ell-1}\epsilon J_{\boldsymbol{\Sigma}^*}(\boldsymbol{\psi}) + \sum_{k=1}^{K-1} \chi_{c^*}^{\ell-1}\epsilon_k J_{\boldsymbol{\Sigma}^*}(\boldsymbol{\psi}^{(k)})\right)$$

$$= \boldsymbol{\Sigma}^* + \underbrace{\chi^{\ell}\epsilon\boldsymbol{\psi} + \sum_{k=1}^{K-1} \chi_{c^*}^{\ell}\epsilon_k \boldsymbol{\psi}^{(k)}}_{= \boldsymbol{E}^{(\ell)}}.$$

$\square$

**Lemma 3.1** (Iterated map). *For all $d \in [D]$, the covariance $\boldsymbol{\Sigma}^{(\ell)} := \mathbb{E}_{\Theta^{1:\ell}, \Xi^{1:\ell}}\left[\mathbf{H}_{:,d}^{(\ell)} \, \mathbf{H}_{:,d}^{(\ell)\top}\right]$ is obtained recursively by the iterated map $\mathcal{C}$ defined by*

$$\Sigma_{\alpha,\alpha'}^{(\ell)} = \underbrace{\sigma^2 \sum_{k=0}^{K-1} c_k \mathbb{E}\left[\left|[\boldsymbol{F}\phi(\mathbf{H}_{:,d})]_k\right|^2\right] \cos\left(\theta_{\alpha,\alpha'}^{(k)}\right) + \sigma_b^2}_{=: \mathcal{C}(\boldsymbol{\Sigma}^{(\ell-1)})_{\alpha,\alpha'}}, \tag{4}$$

*where the expectation is taken over the pre-activations $\mathbf{H}_{:,d} \sim \mathcal{N}(0, \boldsymbol{\Sigma}^{(\ell-1)})$, $\theta_{\alpha,\alpha'}^{(k)} := \frac{2\pi k}{N}(\alpha - \alpha')$ represents the scaled positional difference.*

*Proof.* For simplicity, we introduce $Y_{\alpha,k,\beta,i}$ as follows.

$$H_{\alpha,d}^{(\ell,k)} = \sum_{i=1}^{D} \sum_{\beta=0}^{N-1} \underbrace{F_{\alpha,k}^{\dagger} F_{k,\beta} X_{\beta,i}^{(\ell-1)}}_{=: Y_{\alpha,k,\beta,i}} \left(\Theta_{i,d}^{(\ell,k)} + \left(1 - \left(\delta_{k,0} + \delta_{k,N/2}\right)\right)\sqrt{-1}\Xi_{i,d}^{(\ell,k)}\right).$$

Since the weights are sampled independently, for different $k \neq k'$,

$$\mathbb{E}\left[H_{\alpha,d}^{(\ell,k)} H_{\alpha,d}^{(\ell,k')}\right] = \mathbb{E}\left[H_{\alpha,d}^{(\ell,k)} \overline{H}_{\alpha,d}^{(\ell,k')}\right] = \mathbb{E}\left[\overline{H}_{\alpha,d}^{(\ell,k)} \overline{H}_{\alpha,d}^{(\ell,k')}\right] = 0.$$

From Eq. (3), we have

$$\mathbb{E}_{\Theta^{(\ell)}, \Xi^{(\ell)}}[H_{\alpha,d}^{(\ell)} H_{\alpha',d}^{(\ell)}] = \sum_{k=0}^{K-1} \frac{c_k}{2} \mathbb{E}\left[\left(H_{\alpha,d}^{(\ell,k)} + \overline{H}_{\alpha,d}^{(\ell,k)}\right)\left(H_{\alpha',d}^{(\ell,k)} + \overline{H}_{\alpha',d}^{(\ell,k)}\right)\right] + \mathbb{E}\left[(b_d^{(\ell)})^2\right].$$

First, we calculate the term $k \neq 0, \frac{N}{2}$, where $c_k = 2$.

$$\mathbb{E}\left[\left(H_{\alpha,d}^{(\ell,k)} + \overline{H}_{\alpha,d}^{(\ell,k)}\right)\left(H_{\alpha',d}^{(\ell,k)} + \overline{H}_{\alpha',d}^{(\ell,k)}\right)\right]$$

$$= \sum_{i,i'=1}^{D}\sum_{\beta,\beta'=0}^{N-1} Y_{\alpha,k,\beta,i}Y_{\alpha',k,\beta',i'}\delta_{i,i'}\left(\sigma^2 - \sigma^2\right)/2D$$

$$+ \sum_{i,i'=1}^{D}\sum_{\beta,\beta'=1}^{N} \overline{Y}_{\alpha,k,\beta,i}\overline{Y}_{\alpha',k,\beta',i'}\delta_{i,i'}\left(\sigma^2 - \sigma^2\right)/2D$$

$$+ \sum_{i,i'=1}^{D}\sum_{\beta,\beta'=0}^{N-1} Y_{\alpha,k,\beta,i}\overline{Y}_{\alpha',k,\beta',i'}\delta_{i,i'}\left(\sigma^2 + \sigma^2\right)/2D$$

$$+ \sum_{i,i'=1}^{D}\sum_{\beta,\beta'=0}^{N-1} \overline{Y}_{\alpha,k,\beta,i}Y_{\alpha',k,\beta',i'}\delta_{i,i'}\left(\sigma^2 + \sigma^2\right)/2D$$

$$= \frac{\sigma^2}{D}\sum_{i=1}^{D}\sum_{\beta,\beta'=0}^{N-1} Y_{\alpha,k,\beta,i}\overline{Y}_{\alpha',k,\beta',i} + \overline{Y}_{\alpha,k,\beta,i}Y_{\alpha',k,\beta',i}$$

$$= \frac{\sigma^2}{D}\sum_{i=1}^{D}\underbrace{\left(\sum_{\beta=0}^{N-1}F_{k,\beta}X_{\beta,i}^{(\ell-1)}\right)}_{=\hat{X}_{k,i}^{(\ell-1)}}\overline{\left(\sum_{\beta'=0}^{N-1}F_{k,\beta'}X_{\beta',i}^{(\ell-1)}\right)}\left(F_{\alpha,k}^{\dagger}F_{k,\alpha'} + \overline{F_{\alpha,k}^{\dagger}F_{k,\alpha'}}\right)$$

$$= \frac{2\sigma^2}{D}\sum_{i=1}^{D}\left|\hat{X}_{k,i}^{(\ell-1)}\right|^2\cos\left(\theta_{\alpha,\alpha'}^{(k)}\right),$$

where $\hat{X}_{k,i}^{(\ell-1)}$ is the $k$-th Fourier mode of the representaion $\mathbf{X}_{:,i}^{(\ell-1)}$.

Second, we calculate the terms $k = 0, \frac{N}{2}$, where $c_k = 1$ and $H_{\alpha,d}^{(\ell,k)} = \overline{H}_{\alpha,d}^{(\ell,k)}$.

$$\mathbb{E}\left[\frac{c_k}{2}\left(H_{\alpha,d}^{(\ell,k)} + \overline{H}_{\alpha,d}^{(\ell,k)}\right)\left(H_{\alpha',d}^{(\ell,k)} + \overline{H}_{\alpha',d}^{(\ell,k)}\right)\right] = 2\sum_{i,i'=1}^{D}\sum_{\beta,\beta'=0}^{N-1}\left(Y_{\alpha,k,\beta,i}Y_{\alpha',k,\beta',i'}\right)\delta_{i,i'}\frac{\sigma^2}{2D}$$

$$= \frac{\sigma^2}{D}\sum_{i=1}^{D}\left|\hat{X}_{k,i}^{(\ell-1)}\right|^2\cos\left(\theta_{\alpha,\alpha'}^{(k)}\right).$$

Since the Fourier modes $|\hat{X}_{k,i}^{(\ell-1)}|^2$ are i.i.d. for each hidden dimension $i \in [D]$, we have

$$\mathbb{E}_{\boldsymbol{\Theta}^{0:\ell-1},\boldsymbol{\Xi}^{0:\ell-1},\boldsymbol{b}^{0:\ell-1}}\mathbb{E}_{\boldsymbol{\Theta}^{(\ell)},\boldsymbol{\Xi}^{(\ell)},\boldsymbol{b}^{(\ell)}}\left[H_{\alpha,d}^{(\ell)}H_{\alpha',d}^{(\ell)}\right] = \sigma^2\sum_{k=0}^{K-1}c_k\mathbb{E}_{\boldsymbol{\Theta}^{0:\ell-1},\boldsymbol{\Xi}^{0:\ell-1}}\left[\left|\hat{X}_{k,i}^{(\ell-1)}\right|^2\right]\cos\left(\theta_{\alpha,\alpha'}^{(k)}\right) + \sigma_b^2$$

$$= \sigma^2\sum_{k=0}^{K-1}c_k\mathbb{E}_{\mathbf{H}_{:,d}\sim\mathcal{N}\left(0,\boldsymbol{\Sigma}^{(\ell-1)}\right)}\left[\left|\hat{X}_{k,i}^{(\ell-1)}\right|^2\right]\cos\left(\theta_{\alpha,\alpha'}^{(k)}\right) + \sigma_b^2.$$
$$(16)$$

To obtain a more tractable expression in the subsequent proofs of theorems, we express the iterated map without using Fourier modes as follows.

$$\mathbb{E}_{\boldsymbol{\Theta}^{0:\ell},\boldsymbol{\Xi}^{0:\ell},\boldsymbol{b}^{0:\ell}}\left[H_{\alpha,d}^{(\ell)}H_{\alpha',d}^{(\ell)}\right]$$

$$= \frac{\sigma^2}{N^2}\sum_{k=0}^{K-1}c_k\sum_{\beta,\beta'=0}^{N-1}\cos\left(\theta_{\alpha,\alpha'}^{(k)}\right)\cos\left(\theta_{\beta,\beta'}^{(k)}\right)\mathbb{E}_{\mathbf{H}_{:,d}\sim\mathcal{N}\left(0,\boldsymbol{\Sigma}^{(\ell-1)}\right)}\left[\phi(H_{\beta,d})\phi(H_{\beta',d})\right] + \sigma_b^2. \quad (17)$$

$\square$

**Lemma 3.2** (Exsistance of fixed points). *When a random DCN defined by Eq. (2) has the fixed point $(q^*, c^* = 1)$ for the initial parameters $(\sigma^2, \sigma_b^2)$, then a random simplified FNO defined by Eq. (3) has a fixed point $\Sigma^*$ of the form*

$$\Sigma^* = q^* I_N + q^* c^* (1_N 1_N^\top - I_N) = q^* 1_N 1_N^\top.$$

*Proof.* Using the properties of cosine functions, the following holds.

$$\sigma_b^2 = \frac{1}{N^2} \sum_{k=0}^{K-1} c_k \cos\left(\theta_{\alpha,\alpha'}^{(k)}\right) \underbrace{\sum_{\beta,\beta'=0}^{N-1} \cos\left(\theta_{\beta,\beta'}^{(k)}\right)}_{=N^2 \delta_{k,0}} \sigma_b^2. \tag{18}$$

Then, the following holds for all $\alpha, \alpha' \in [N]$.

$$[\mathcal{C}(\Sigma^*)]_{\alpha,\alpha'} = \frac{1}{N^2} \sum_{k=0}^{K-1} c_k \cos\left(\theta_{\alpha,\alpha'}^{(k)}\right) \sum_{\beta,\beta'=0}^{N-1} \cos\left(\theta_{\beta,\beta'}^{(k)}\right) \left(\sigma^2 \mathbb{E}_{H_{:,d}\sim\mathcal{N}(0,\Sigma^*)}\left[\phi(H_{\beta,d})\phi(H_{\beta',d})\right] + \sigma_b^2\right).$$

When the DCN is applied pointwise to each spatial representation $H_{\beta,:}$, $\beta \in [N]$, the iterated map of the random DCN (Poole et al., 2016) is given by $\mathcal{C}_{\text{DCN}}(\Sigma) := \sigma^2 \mathbb{E}_{H_{:,d}\sim\mathcal{N}(0,\Sigma)}\left[\phi(H_{\beta,d})\phi(H_{\beta',d})\right] + \sigma_b^2$. Since the covariance $\Sigma^*$ is a fixed point with respect to the iterative map of the DCN, *i.e.* $\mathcal{C}_{\text{DCN}}(\Sigma^*) = \Sigma^*$, the following holds.

$$[\mathcal{C}(\Sigma^*)]_{\alpha,\alpha'} = \frac{1}{N^2} \sum_{k=0}^{K-1} c_k \cos\left(\theta_{\alpha,\alpha'}^{(k)}\right) \underbrace{\sum_{\beta,\beta'=0}^{N-1} \cos\left(\theta_{\beta,\beta'}^{(k)}\right)}_{=N^2 \delta_{k,0}} \underbrace{q^*(\delta_{\beta,\beta'} + (1 - \delta_{\beta,\beta'})c^*)}_{=q^*} = q^*.$$

Thus, we confirm that the covariance $\Sigma^*$ satisfies the definition of a fixed point in the map $\mathcal{C}$, *i.e.* $\Sigma^* = \mathcal{C}(\Sigma^*)$. This means that the fixed point for the iterated map $\Sigma^*$ of the DCN also serves as a fixed point for the iterated map of the simplified FNO. □

**Lemma A.1.** *Let $\Sigma^*$ be the fixed point of the form in Eq. (5). Suppose that the symmetric perturbation $E \in \mathbb{R}^{N \times N}$, where $E_{\beta,\beta} = E_{\beta',\beta'}$ and $E_{\beta,\beta'}$ are non-zero for some $\beta, \beta' \in [N]$, $\beta \neq \beta$, and all other elements are zero. Then, we have*

$$\sigma^2 \mathbb{E}_{H_{:,d}\sim\mathcal{N}(0,\Sigma^*+E)}\left[\phi(H_{\beta,d})^2\right] + \sigma_b^2 = q^* + E_{\beta,\beta}\chi_{q^*} + \mathcal{O}\left(|E_{\beta,\beta}|^2\right), \tag{19}$$

$$\sigma^2 \mathbb{E}_{H_{:,d}\sim\mathcal{N}(0,\Sigma^*+E)}\left[\phi(H_{\beta,d})\phi(H_{\beta',d})\right] + \sigma_b^2 = q^*c^* + E_{\beta,\beta}\chi_\kappa + E_{\beta,\beta'}\chi_{c^*} + \mathcal{O}\left(|E_{\beta,\beta'}|^2\right), \tag{20}$$

*where $\chi_{q^*}$, $\chi_{c^*}$, and $\chi_\kappa$ are the constants defined by*

$$\chi_{q^*} := \sigma^2 \mathbb{E}_{H_{:,d}\sim\mathcal{N}(0,\Sigma^*)}\left[\phi'^2(H_{\beta,d}) + \phi''(H_{\beta,d})\phi(H_{\beta,d})\right],$$

$$\chi_{c^*} := \sigma^2 \mathbb{E}_{H_{:,d}\sim\mathcal{N}(0,\Sigma^*)}\left[\phi'(H_{\beta,d})\phi'(H_{\beta',d})\right],$$

$$\chi_\kappa := \frac{\sigma^2}{2} \mathbb{E}\left[\phi''(H_{\alpha,d})\phi(H_{\alpha',d}) + \phi(H_{\alpha,d})\phi''(H_{\alpha',d}) + 2c^*\phi'(H_{\alpha,d})\phi'(H_{\alpha',d})\right].$$

*Proof.* Equation (19) is obviously shown by the result of Section 2 in (Poole et al., 2016) and Section 3 in (Schoenholz et al., 2016). We prove Eq. (20) with reference to the results of (Schoenholz et al., 2016).

$$\sigma^2 \mathbb{E}_{H_{:,d}\sim\mathcal{N}(0,\Sigma^*+E)}\left[\phi(H_{\beta,d})\phi(H_{\beta',d})\right] + \sigma_b^2 = \sigma^2 \int \mathcal{D}z_1 \mathcal{D}z_2 \phi(u_1)\phi(u_2) + \sigma_b^2,$$

where $\int \mathcal{D}\,dz = \frac{1}{\sqrt{2\pi}} \int dz e^{-\frac{1}{2}z^2}$ is the measure for a standard Gaussian distribution, $u_1 = \sqrt{q}z_1$, $u_2 = \sqrt{q}\left(c_{\beta,\beta'}z_1 + \sqrt{1 - c_{\beta,\beta'}^2}z_2\right)$, $q = q^* + E_{\beta,\beta}$ and $c_{\beta,\beta'} = c^* + \frac{E_{\beta,\beta'}}{q}$.

We consider the case where $c^* < 1$ and $c^* = 1$ separately. Later, we will show that the two results agree with each other. First, we consider the case where $c^* < 1$. Using Taylor expansion, we can approximate $\phi(u_1)$ and $\phi(u_2)$ as follows. For the simplicity, we assume $\mathcal{O}(|E_{\beta,\beta}|) = \mathcal{O}(|E_{\beta,\beta'}|)$.

$$\phi(u_1) = \phi(u_1^*) + \frac{1}{2}\frac{E_{\beta,\beta}}{\sqrt{q^*}}z_1\phi'(u_1^*) + \mathcal{O}(|E_{\beta,\beta}|^2),$$

$$\phi(u_2) = \phi(u_2^*) + \frac{E_{\beta,\beta'}}{\sqrt{q^*}}\left(z_1 - \frac{c^*}{\sqrt{1-(c^*)^2}}z_2\right)\phi'(u_2^*) + \frac{E_{\beta,\beta}}{2\sqrt{q^*}}(c^*z_1 + \sqrt{1-(c^*)^2}z_2)\phi'(u_2^*) + \mathcal{O}(|E_{\beta,\beta}|^2),$$

where $u_1^* = \sqrt{q^*}z_1$ and $u_2^* = \sqrt{q^*}(c^*z_1 + \sqrt{1-(c^*)^2}z_2)$.

Thus, we have

$$\sigma^2\int\mathcal{D}z_1\mathcal{D}z_2\phi(u_1)\phi(u_2) + \sigma_b^2$$

$$= \underbrace{\sigma^2\int\mathcal{D}z_1\mathcal{D}z_2\phi(u_1^*)\phi(u_2^*) + \sigma_b^2}_{=q^*c^*} + \sigma^2\int\mathcal{D}z_1\mathcal{D}z_2\frac{E_{\beta,\beta'}}{\sqrt{q^*}}\left(z_1 - \frac{c^*}{\sqrt{1-(c^*)^2}}z_2\right)\phi(u_1^*)\phi'(u_2^*)$$

$$+ \sigma^2\int\mathcal{D}z_1\mathcal{D}z_2\frac{1}{2}\frac{E_{\beta,\beta}}{\sqrt{q^*}}(c^*z_1 + \sqrt{1-(c^*)^2}z_2)\phi(u_1^*)\phi'(u_2^*) + \sigma^2\int\mathcal{D}z_1\mathcal{D}z_2\frac{1}{2}\frac{E_{\beta,\beta}}{\sqrt{q^*}}z_1\phi'(u_1^*)\phi(u_2^*) + \mathcal{O}(|E_{\beta,\beta}^2|).$$

The results of the second term are obtained from the transformation of equations 36 to 39 in Appendix 7.2 of (Schoenholz et al., 2016).

$$\sigma^2\frac{E_{\beta,\beta'}}{\sqrt{q^*}}\int\mathcal{D}z_1\mathcal{D}z_2\left(z_1 - \frac{c^*}{\sqrt{1-(c^*)^2}}z_2\right)\phi(u_1^*)\phi'(u_2^*) = E_{\beta,\beta'}\underbrace{\sigma^2\int\mathcal{D}z_1\mathcal{D}z_2\phi'(u_1^*)\phi'(u_2^*)}_{=\chi_{c^*}}.$$

Utilizing the identity, $\int\mathcal{D}zzf(z) = \int\mathcal{D}zf'(z)$, we obtain the third term as follows.

$$\sigma^2\int\mathcal{D}z_1\mathcal{D}z_2\frac{1}{2}\frac{E_{\beta,\beta}}{\sqrt{q^*}}(c^*z_1 + \sqrt{1-(c^*)^2}z_2)\phi(u_1^*)\phi'(u_2^*)$$

$$= \sigma^2\frac{1}{2}E_{\beta,\beta}\int\mathcal{D}z_1\mathcal{D}z_2\left(c^*\left(\phi'(u_1^*)\phi'(u_2^*) + c^*\phi(u_1^*)\phi''(u_2^*)\right) + (1-(c^*)^2)\phi(u_1^*)\phi''(u_2^*)\right)$$

$$= \sigma^2\frac{1}{2}E_{\beta,\beta}\int\mathcal{D}z_1\mathcal{D}z_2\left(c^*\phi'(u_1^*)\phi'(u_2^*) + \phi(u_1^*)\phi''(u_2^*)\right).$$

The last term is calculated as follows.

$$\sigma^2\int\mathcal{D}z_1\mathcal{D}z_2\frac{1}{2}\frac{E_{\beta,\beta}}{\sqrt{q^*}}z_1\phi'(u_1^*)\phi(u_2^*) = \sigma^2\frac{1}{2}E_{\beta,\beta}\int\mathcal{D}z_1\mathcal{D}z_2(\phi''(u_1^*)\phi(u_2^*) + c^*\phi'(u_1^*)\phi'(u_2^*)).$$

Summing the last two terms, we obtain the term $E_{\beta,\beta}\chi_\kappa$.

Next, we consider the case where $c^* = 1$. As with the discussion of (Schoenholz et al., 2016), the perturbed correlation is defined by $c_{\beta,\beta'} = c^* - \frac{E_{\beta,\beta'}}{q}$ where $E_{\beta,\beta'} > 0$ and then $\phi(u_2)$ is expanded as follows.

$$\phi(u_2) = \phi(u_2^*) + \left(\sqrt{2\frac{E_{\beta,\beta'}}{q^*}}z_2 - \frac{E_{\beta,\beta'}}{\sqrt{q^*}}z_1\right)\phi'(u_2^*) + E_{\beta,\beta'}z_2^2\phi''(u_2^*) + \frac{E_{\beta,\beta}}{2\sqrt{q^*}}z_1\phi'(u_2^*) + \mathcal{O}(|E_{\beta,\beta'}|^{3/2}).$$

Thus, we have

$$\sigma^2 \int \mathcal{D}z_1 \mathcal{D}z_2 \phi(u_1)\phi(u_2) + \sigma_b^2$$

$$= \underbrace{\sigma^2 \int \mathcal{D}z_1 \mathcal{D}z_2 \phi(u_1^*)\phi(u_2^*) + \sigma_b^2}_{=q^* c^*} + \sigma^2 \int \mathcal{D}z_1 \mathcal{D}z_2 \sqrt{2\frac{E_{\beta,\beta'}}{q^*}} z_2 \phi(u_1^*)\phi'(u_2^*)$$

$$- \sigma^2 \int \mathcal{D}z_1 \mathcal{D}z_2 \frac{E_{\beta,\beta'}}{\sqrt{q^*}} z_1 \phi(u_1^*)\phi'(u_2^*) + \sigma^2 \int \mathcal{D}z_1 \mathcal{D}z_2 E_{\beta,\beta'} z_2^2 \phi(u_1^*)\phi''(u_2^*)$$

$$+ \sigma^2 \int \mathcal{D}z_1 \mathcal{D}z_2 \frac{E_{\beta,\beta}}{2\sqrt{q^*}} z_1 \phi(u_1^*)\phi'(u_2^*) + \sigma^2 \int \mathcal{D}z_1 \mathcal{D}z_2 \frac{1}{2}\frac{E_{\beta,\beta}}{\sqrt{q^*}} z_1 \phi'(u_1^*)\phi(u_2^*) + \mathcal{O}(|E_{\beta,\beta}^{3/2}|).$$

$$(21)$$

Using the fact that $u_2^* = u_1^*$ and $u_2^*$ is independent of $z_2$,

$$\sigma^2 \int \mathcal{D}z_1 \mathcal{D}z_2 \sqrt{2\frac{E_{\beta,\beta'}}{q^*}} z_2 \phi(u_1^*)\phi'(u_2^*) = \sigma^2 \int \mathcal{D}z_1 \sqrt{2\frac{E_{\beta,\beta'}}{q^*}} \phi(u_1^*)\phi'(u_2^*) \underbrace{\left(\int \mathcal{D}z_2 z_2\right)}_{=0}.$$

$$\sigma^2 \int \mathcal{D}z_1 \mathcal{D}z_2 \frac{E_{\beta,\beta'}}{\sqrt{q^*}} z_1 \phi(u_1^*)\phi'(u_2^*) = \sigma^2 \int \mathcal{D}z_1 \frac{E_{\beta,\beta'}}{\sqrt{q^*}} z_1 \phi(u_1^*)\phi'(u_1^*)$$

$$= \sigma^2 E_{\beta,\beta'} \int \mathcal{D}z_1 (\phi'(u_1^*)^2 + \phi(u_1^*)\phi''(u_1^*)).$$

$$\sigma^2 \int \mathcal{D}z_1 \mathcal{D}z_2 E_{\beta,\beta'} z_2^2 \phi(u_1^*)\phi''(u_2^*) = \sigma^2 E_{\beta,\beta'} \int \mathcal{D}z_1 \phi(u_1^*)\phi''(u_1^*) \underbrace{\left(\int \mathcal{D}z_2 z_2^2\right)}_{=1}.$$

$$\sigma^2 \int \mathcal{D}z_1 \mathcal{D}z_2 \frac{E_{\beta,\beta}}{2\sqrt{q^*}} z_1 \phi(u_1^*)\phi'(u_2^*) = \sigma^2 \int \mathcal{D}z_1 \frac{E_{\beta,\beta}}{2\sqrt{q^*}} z_1 \phi(u_1^*)\phi'(u_1^*)$$

$$= \sigma^2 \frac{E_{\beta,\beta}}{2} \int \mathcal{D}z_1 (\phi'(u_1^*)^2 + \phi(u_1^*)\phi''(u_1^*)).$$

$$\sigma^2 \int \mathcal{D}z_1 \mathcal{D}z_2 \frac{1}{2}\frac{E_{\beta,\beta}}{\sqrt{q^*}} z_1 \phi'(u_1^*)\phi(u_2^*) = \sigma^2 \int \mathcal{D}z_1 \frac{1}{2}\frac{E_{\beta,\beta}}{\sqrt{q^*}} z_1 \phi'(u_1^*)\phi(u_1^*)$$

$$= \sigma^2 \frac{E_{\beta,\beta}}{2} \int \mathcal{D}z_1 (\phi'(u_1^*)^2 + \phi(u_1^*)\phi''(u_1^*)).$$

Substituting these facts into Eq. (21), we obtain

$$\sigma^2 \int \mathcal{D}z_1 \mathcal{D}z_2 \phi(u_1)\phi(u_2) + \sigma_b^2$$

$$\approx q^* c^* - \sigma^2 E_{\beta,\beta'} \int \mathcal{D}z_1 \phi'(u_1^*)^2 + \sigma^2 E_{\beta,\beta} \int \mathcal{D}z_1 (\phi'(u_1^*)^2 + \phi(u_1^*)\phi''(u_1^*))$$

The above result agrees with that obtained by substituting $c^* = 1$ for the result obtained when the case $c^* < 1$.

$$\square$$

**Lemma A.2.** *The Jacobian matrix $J(\boldsymbol{\Sigma}^*)$ of the iterated map $\mathcal{C}$ defined in Eq. (14) is obtained as follows.*

$$[J(\boldsymbol{\Sigma}^*)]_{(\alpha,\alpha'),(\beta,\beta')} = \frac{1}{N^2} \sum_{k'=0}^{K-1} c_{k'} \cos\left(\theta_{\alpha,\alpha'}^{(k')}\right) \cos\left(\theta_{\beta,\beta'}^{(k')}\right) (\delta_{\beta,\beta'}(\chi_{q^*} - \chi_\kappa + \delta_{k',0} N\chi_\kappa) + (1 - \delta_{\beta,\beta'})\chi_{c_*})).$$

*Furthermore, the rank of the Jacobian matrix $J(\boldsymbol{\Sigma}^*) \in \mathbb{R}^{N^2 \times N^2}$ is at most $K$.*

*Proof.* Let some semi-positive definite matrix $\boldsymbol{E} \in \mathbb{R}^{N \times N}$ be a deviation from the fixed point $\boldsymbol{\Sigma}^*$. From Lemma 3.1 and Lemma A.1, we have

$$
[\mathcal{C}(\boldsymbol{\Sigma}^* + \boldsymbol{E})]_{\alpha,\alpha'}
$$

$$
= \frac{1}{N^2} \sum_{k'=0}^{K-1} c_{k'} \cos\left(\theta_{\alpha,\alpha'}^{(k')}\right) \sum_{\beta,\beta'=0}^{N-1} \cos\left(\theta_{\beta,\beta'}^{(k')}\right) \left(\sigma^2 \mathbb{E}_{\mathbf{H}_{:,d} \sim \mathcal{N}(0,\boldsymbol{\Sigma}^*+\boldsymbol{E})} \left[\phi(H_{\gamma,d})\phi(H_{\gamma',d})\right] + \sigma_b^2\right)
$$

$$
\approx \frac{1}{N^2} \sum_{k'=0}^{K-1} c_{k'} \cos\left(\theta_{\alpha,\alpha'}^{(k')}\right) \sum_{\beta,\beta'=0}^{N-1} \cos\left(\theta_{\beta,\beta'}^{(k')}\right) \left(\delta_{\beta,\beta'}(q^* + E_{\beta,\beta}\chi_{q^*}) + (1 - \delta_{\beta,\beta'})(q^* c^* + E_{\beta,\beta}\chi_\kappa + E_{\beta,\beta'}\chi_{c_*})\right).
$$

$$(22)$$

Note that Eq. (22) is obtained by neglecting higher order terms in Eqs. (19) and (20).

By the definition of the fixed point,

$$
\Sigma_{\alpha,\alpha'}^* = \frac{1}{N^2} \sum_{k'=0}^{K-1} c_{k'} \cos\left(\theta_{\alpha,\alpha'}^{(k')}\right) \sum_{\beta,\beta'=0}^{N-1} \cos\left(\theta_{\beta,\beta'}^{(k')}\right) q^*(\delta_{\beta,\beta'} + (1 - \delta_{\beta,\beta'})c^*).
$$

By substituting the fact into Eq. (22), we obtain

$$
[\mathcal{C}(\boldsymbol{\Sigma}^* + \boldsymbol{E}) - \boldsymbol{\Sigma}^*]_{\alpha,\alpha'}
$$

$$
\approx \frac{1}{N^2} \sum_{k'=0}^{K-1} c_{k'} \cos\left(\theta_{\alpha,\alpha'}^{(k')}\right) \sum_{\beta,\beta'=0}^{N-1} \cos\left(\theta_{\beta,\beta'}^{(k')}\right) \left(\delta_{\beta,\beta'} E_{\beta,\beta}\chi_{q^*}) + (1 - \delta_{\beta,\beta'})(E_{\beta,\beta}\chi_\kappa + E_{\beta,\beta'}\chi_{c_*})\right)
$$

$$(23)$$

$$
= \sum_{\beta,\beta'=0}^{N-1} \underbrace{\left[\frac{1}{N^2} \sum_{k'=0}^{K-1} c_{k'} \cos\left(\theta_{\alpha,\alpha'}^{(k')}\right) \cos\left(\theta_{\beta,\beta'}^{(k')}\right) \left(\delta_{\beta,\beta'}(\chi_{q^*} - \chi_\kappa + \delta_{k',0} N\chi_\kappa) + (1 - \delta_{\beta,\beta'})\chi_{c_*}\right)\right]}_{=[J(\boldsymbol{\Sigma}^*)]_{(\alpha,\alpha'),(\beta,\beta')}} E_{\beta,\beta'}.
$$

The last equation can be rewritten using the matrix calculation as follows.

$$
\mathcal{C}(\boldsymbol{\Sigma}^* + \boldsymbol{E}) - \boldsymbol{\Sigma}^* \approx \mathrm{mat}(J(\boldsymbol{\Sigma}^*) \, \mathrm{vec}(\boldsymbol{E})).
$$

Since $J(\boldsymbol{\Sigma}^*)$ is the first-order coefficient to the deviaction $\boldsymbol{E}$, $J(\boldsymbol{\Sigma}^*)$ is exactly the Jacobian matrix of the iterated map $\mathcal{C}$.

Furthermore, the Jacobian matrix can be decomposed to two matricies $\boldsymbol{A} \in \mathbb{R}^{N^2 \times K}$ and $\boldsymbol{B} \in \mathbb{R}^{K \times N^2}$ as follows.

$$
[J(\boldsymbol{\Sigma}^*)]_{(\alpha,\alpha'),(\beta,\beta')} = \sum_{k'=0}^{K-1} A_{(\alpha,\alpha'),k'} B_{k',(\beta,\beta')}, \ A_{(\alpha,\alpha'),k'} := \cos\left(\theta_{\alpha,\alpha'}^{(k')}\right),
$$

$$
B_{k',(\beta,\beta')} := \frac{1}{N^2} c_{k'} \cos\left(\theta_{\beta,\beta'}^{(k')}\right) \left(\delta_{\beta,\beta'}(\chi_{q^*} - \chi_\kappa + \delta_{k',0} N\chi_\kappa) + (1 - \delta_{\beta,\beta'})\chi_{c_*}\right).
$$

Therefore, the rank of the Jacobian matrix is at most $K$. $\qquad\square$

**Lemma A.3.** *Let $\boldsymbol{\Sigma}^*$ be the fixed point of the form in Eq.* (5) *and $\boldsymbol{E}^{(k)}$ be the perturbation expressed as*

$$
E_{\beta,\beta'}^{(k)} = \epsilon_k \psi_{\beta,\beta'}^{(k)}, \quad \psi_{\beta,\beta'}^{(k)} := \left[\cos\left(\theta_{\beta,\beta'}^{(k)}\right) - \left(\sum_{s=0}^{K-1} c_s\right)^{-1} \sum_{s=0}^{K-1} c_s \cos\left(\theta_{\beta,\beta'}^{(s)}\right)\right],
$$

*where $\epsilon_k$ denotes the scale of the perturbation, assumed to be sufficiently small.*

*Then, we have, for all $k \in [K] \backslash \{0\}$,*

$$
\mathcal{C}(\boldsymbol{\Sigma}^* + \boldsymbol{E}^{(k)}) \approx \boldsymbol{\Sigma}^* + \chi_{c^*} \boldsymbol{E}^{(k)}.
$$

*Proof.* From Eq. (23), we have

$$[\mathcal{C}(\mathbf{\Sigma}^* + \mathbf{E}^{(k)})]_{\alpha,\alpha'} \approx \frac{1}{N^2} \sum_{k'=0}^{K-1} c_{k'} \cos\left(\theta_{\alpha,\alpha'}^{(k')}\right) \sum_{\beta=0}^{N-1} (q^* + E_{\beta,\beta}^{(k)} \chi_{q^*})$$

$$+ \frac{1}{N^2} \sum_{k'=0}^{K-1} c_{k'} \cos\left(\theta_{\alpha,\alpha'}^{(k')}\right) \sum_{\beta=0}^{N-1} \sum_{\beta \neq \beta'} \cos\left(\theta_{\beta,\beta'}^{(k')}\right) (q^* c^* + E_{\beta,\beta}^{(k)} \chi_\kappa + E_{\beta,\beta'}^{(k)} \chi_{c^*}).$$

From the definition of the fixed point, we obtain

$$\Sigma_{\alpha,\alpha'}^* = \frac{1}{N^2} \sum_{k'=0}^{K-1} c_{k'} \cos\left(\theta_{\alpha,\alpha'}^{(k')}\right) \sum_{\beta,\beta'=0}^{N-1} \cos\left(\theta_{\beta,\beta'}^{(k')}\right) q^* (\delta_{\beta,\beta'} + (1 - \delta_{\beta,\beta'})c^*).$$

By combining the above two results, we have

$$[\mathcal{C}(\mathbf{\Sigma}^* + \mathbf{E}^{(k)}) - \mathbf{\Sigma}^*]_{\alpha,\alpha'} \approx \frac{1}{N^2} \sum_{k'=0}^{K-1} c_{k'} \cos\left(\theta_{\alpha,\alpha'}^{(k')}\right) \sum_{\beta=0}^{N-1} E_{\beta,\beta}^{(k)} \chi_{q^*}$$

$$+ \frac{1}{N^2} \sum_{k'=0}^{K-1} c_{k'} \cos\left(\theta_{\alpha,\alpha'}^{(k')}\right) \sum_{\beta=0}^{N-1} \sum_{\beta \neq \beta'} \cos\left(\theta_{\beta,\beta'}^{(k')}\right) (E_{\beta,\beta}^{(k)} \chi_\kappa + E_{\beta,\beta'}^{(k)} \chi_{c^*}).$$

Since $\cos(\theta_{\beta,\beta}^{(k)}) = 1$ for all $k \in [K]$, we have $\psi_{\beta,\beta}^{(k)} = 0$ and $E_{\beta,\beta}^{(k)} = 0$. Thus, only the term with $E_{\beta,\beta'}^{(k)}$ remains.

$$[\mathcal{C}(\mathbf{\Sigma}^* + \mathbf{E}^{(k)}) - \mathbf{\Sigma}^*]_{\alpha,\alpha'} \approx \epsilon_k \chi_{c^*} \frac{1}{N^2} \left( \sum_{k'=0}^{K-1} c_{k'} \cos\left(\theta_{\alpha,\alpha'}^{(k')}\right) \sum_{\beta \neq \beta'} \cos\left(\theta_{\beta,\beta'}^{(k')}\right) \cos\left(\theta_{\beta,\beta'}^{(k)}\right) \quad (24)$$

$$- \left( \sum_{s=0}^{K-1} c_s \right)^{-1} \sum_{k'=0}^{K-1} c_{k'} \cos\left(\theta_{\alpha,\alpha'}^{(k')}\right) \sum_{\beta \neq \beta'} \cos\left(\theta_{\beta,\beta'}^{(k')}\right) \sum_{s=0}^{K-1} c_s \cos\left(\theta_{\beta,\beta'}^{(s)}\right) \right).$$

$$(25)$$

Given the orthogonality of the cosine functions, we have

$$\sum_{\substack{\beta,\beta'=0 \\ \beta \neq \beta'}}^{N-1} c_{k'} \cos\left(\theta_{\beta,\beta'}^{(k')}\right) \cos\left(\theta_{\beta,\beta'}^{(k)}\right) = \begin{cases} N^2 - c_{k'} N & (k' = k) \\ -c_{k'} N & \text{(otherwise)}. \end{cases} \quad (26)$$

Utilizing Eq. (26) leads to the following two facts:

$$\sum_{k'=0}^{K-1} c_{k'} \cos\left(\theta_{\alpha,\alpha'}^{(k')}\right) \sum_{\beta \neq \beta'} \cos\left(\theta_{\beta,\beta'}^{(k')}\right) \cos\left(\theta_{\beta,\beta'}^{(k)}\right) = N^2 \cos\left(\theta_{\alpha,\alpha'}^{(k)}\right) - N \sum_{k'=0}^{K-1} c_{k'} \cos\left(\theta_{\alpha,\alpha'}^{(k')}\right).$$

$$\sum_{k'=0}^{K-1} c_{k'} \cos\left(\theta_{\alpha,\alpha'}^{(k')}\right) \sum_{\beta \neq \beta'} \cos\left(\theta_{\beta,\beta'}^{(k')}\right) \sum_{s=0}^{K-1} c_s \cos\left(\theta_{\beta,\beta'}^{(s)}\right) = N^2 \sum_{k'=0}^{K-1} \cos\left(\theta_{\alpha,\alpha'}^{(k')}\right) - N \left( \sum_{s=0}^{K-1} c_s \right) \sum_{k'=0}^{K-1} c_{k'} \cos\left(\theta_{\alpha,\alpha'}^{(k')}\right).$$

By substituting these facts into Eq. (24), we obtain

$$[\mathcal{C}(\boldsymbol{\Sigma}^* + \boldsymbol{E}^{(k)}) - \boldsymbol{\Sigma}^*]_{\alpha,\alpha'}$$

$$\approx \chi_{c^*} \epsilon_k \frac{1}{N^2} \left( N^2 \cos\left(\theta_{\alpha,\alpha'}^{(k)}\right) - N \sum_{k'=0}^{K-1} \cancel{c_{k'} \cos\left(\theta_{\alpha,\alpha'}^{(k')}\right)} \right.$$

$$\left. - \left( \sum_{s=0}^{K-1} c_s \right)^{-1} N^2 \sum_{k'=0}^{K-1} \cos\left(\theta_{\alpha,\alpha'}^{(k')}\right) + N \sum_{k'=0}^{K-1} \cancel{c_{k'} \cos\left(\theta_{\alpha,\alpha'}^{(k')}\right)} \right)$$

$$= \chi_{c^*} \epsilon_k \underbrace{\left[ \cos\left(\theta_{\alpha,\alpha'}^{(k)}\right) - \left( \sum_{s=0}^{K-1} c_s \right)^{-1} \sum_{k'=0}^{K-1} \cos\left(\theta_{\alpha,\alpha'}^{(k')}\right) \right]}_{=\psi_{\alpha,\alpha'}^{(k)}}.$$

This completes the proof. □

**Lemma A.4.** *Let $\boldsymbol{\Sigma}^*$ be the fixed point of the form in Eq. (5) and $\boldsymbol{E}$ be the perturbation expressed as*

$$E_{\beta,\beta'} = \epsilon \psi_{\beta,\beta'}, \quad \psi_{\beta,\beta'} := \left[ 1 - \frac{1}{N} \left( \frac{\chi_\kappa + \chi_{c^*} - \chi_{q^*}}{\chi_\kappa} \right) \sum_{s=0}^{K-1} c_s \cos\left(\theta_{\beta,\beta'}^{(s)}\right) \right],$$

*where $\epsilon$ denotes the scale of the perturbation, assumed to be sufficiently small, $\mathbf{1}_N \mathbf{1}_N^\top$ is all-one matrix with size $N \times N$.*

*Then, we have*

$$\mathcal{C}(\boldsymbol{\Sigma}^* + \boldsymbol{E}) \approx \boldsymbol{\Sigma}^* + \underbrace{\left( \frac{\sum_{s=0}^{K-1} c_s}{N} \chi_{q^*} + \left( 1 - \frac{\sum_{s=0}^{K-1} c_s}{N} \right) (\chi_\kappa + \chi_{c^*}) \right)}_{:=\chi} \boldsymbol{E}.$$

*Proof.* For simplicity, we introduce $x := \frac{1}{N} \left( \frac{\chi_\kappa + \chi_{c^*} - \chi_{q^*}}{\chi_\kappa} \right) \left( \sum_{s=0}^{K-1} c_s \right)$ as follows.

$$\mathcal{C}(\boldsymbol{\Sigma}^* + \boldsymbol{E}) \approx \boldsymbol{\Sigma}^* + \left( \frac{\sum_{s=0}^{K-1} c_s}{N} \chi_{q^*} + \left( 1 - \frac{\sum_{s=0}^{K-1} c_s}{N} \right) (\chi_\kappa + \chi_{c^*}) \right) \boldsymbol{E}$$

$$= \boldsymbol{\Sigma}^* + \left[ \underbrace{\left( 1 - \frac{1}{N} \left( \frac{\chi_\kappa + \chi_{c^*} - \chi_{q^*}}{\chi_\kappa} \right) \left( \sum_{s=0}^{K-1} c_s \right) \right)}_{=:x} \chi_\kappa + \chi_{c^*} \right] \boldsymbol{E}. \quad (27)$$

From Eq. (23), we have

$$[\mathcal{C}(\boldsymbol{\Sigma}^* + \boldsymbol{E}) - \boldsymbol{\Sigma}^*]_{\alpha,\alpha'}$$

$$\approx \frac{1}{N^2} \sum_{k'=0}^{K-1} c_{k'} \cos\left(\theta_{\alpha,\alpha'}^{(k')}\right) \sum_{\beta=0}^{N-1} \left( E_{\beta,\beta} \chi_{q^*} + \sum_{\beta \neq \beta'} \cos\left(\theta_{\beta,\beta'}^{(k')}\right) (E_{\beta,\beta} \chi_\kappa + E_{\beta,\beta'} \chi_{c^*}) \right). \quad (28)$$

Our goal is to derive Eq. (27) from Eq. (28). The following results are useful for the computation of each term of Eq. (28).

$$\forall \beta, \beta' \in [N], \ \beta \neq \beta', \quad E_{\beta,\beta} = \epsilon(1 - x), \quad E_{\beta,\beta'} = \epsilon \left( 1 - x \left( \sum_{s=0}^{K-1} c_s \right)^{-1} \sum_{s=0}^{K-1} c_s \cos\left(\theta_{\beta,\beta'}^{(s)}\right) \right).$$

$$\sum_{\beta=0}^{N-1} \sum_{\beta \neq \beta'} \cos\left(\theta_{\beta,\beta'}^{(k')}\right) = \begin{cases} N^2 - N & (k' = 0) \\ -N & (\text{otherwise}) \end{cases}.$$

$$\sum_{\substack{\beta,\beta'=0 \\ \beta \neq \beta'}}^{N-1} c_{k'} \cos\left(\theta_{\beta,\beta'}^{(k')}\right) \cos\left(\theta_{\beta,\beta'}^{(k)}\right) = \begin{cases} N^2 - c_{k'} N & (k' = k) \\ -c_{k'} N & (\text{otherwise}) \end{cases}.$$

Using the above results, the first and second terms of Eq. (28) are calculated as follows.

$$\frac{1}{N^2} \sum_{k'=0}^{K-1} c_{k'} \cos\left(\theta_{\alpha,\alpha'}^{(k')}\right) \sum_{\beta=0}^{N-1} E_{\beta,\beta} \chi_{q^*} = \epsilon(1-x)\chi_{q^*} \frac{1}{N} \sum_{k'=0}^{K-1} c_{k'} \cos\left(\theta_{\alpha,\alpha'}^{(k')}\right).$$

$$\frac{1}{N^2} \sum_{k'=0}^{K-1} c_{k'} \cos\left(\theta_{\alpha,\alpha'}^{(k')}\right) \sum_{\beta=0}^{N-1} \sum_{\beta \neq \beta'} \cos\left(\theta_{\beta,\beta'}^{(k')}\right) E_{\beta,\beta} \chi_{\kappa} = \epsilon(1-x)\chi_{\kappa} - \epsilon(1-x)\chi_{\kappa} \frac{1}{N} \sum_{k'=0}^{K-1} c_{k'} \cos\left(\theta_{\alpha,\alpha'}^{(k')}\right).$$

The third term of Eq. (28) is obtained as follows.

$$\frac{1}{N^2} \sum_{k'=0}^{K-1} c_{k'} \cos\left(\theta_{\alpha,\alpha'}^{(k')}\right) \sum_{\beta=0}^{N-1} \sum_{\beta \neq \beta'} \cos\left(\theta_{\beta,\beta'}^{(k')}\right) E_{\beta,\beta'} \chi_{c^*}$$

$$= \frac{1}{N^2} \sum_{k'=0}^{K-1} c_{k'} \cos\left(\theta_{\alpha,\alpha'}^{(k')}\right) \sum_{\beta=0}^{N-1} \sum_{\beta \neq \beta'} \cos\left(\theta_{\beta,\beta'}^{(k')}\right) \epsilon \chi_{c^*}$$

$$- x \left(\sum_{s=0}^{K-1} c_s\right)^{-1} \frac{1}{N^2} \sum_{k'=0}^{K-1} c_{k'} \cos\left(\theta_{\alpha,\alpha'}^{(k')}\right) \sum_{s=0}^{K-1} \sum_{\beta=0}^{N-1} \sum_{\beta \neq \beta'} c_s \cos\left(\theta_{\beta,\beta'}^{(s)}\right) \cos\left(\theta_{\beta,\beta'}^{(k')}\right) \epsilon \chi_{c^*}$$

$$= \epsilon \chi_{c^*} - \epsilon \chi_{c^*} \frac{1}{N} \sum_{k'=0}^{K-1} c_{k'} \cos\left(\theta_{\alpha,\alpha'}^{(k')}\right)$$

$$- \epsilon \chi_{c^*} x \left(\sum_{s=0}^{K-1} c_s\right)^{-1} \sum_{k'=0}^{K-1} c_{k'} \cos\left(\theta_{\alpha,\alpha'}^{(k')}\right) + \epsilon \chi_{c^*} x \frac{1}{N} \sum_{k'=0}^{K-1} c_{k'} \cos\left(\theta_{\alpha,\alpha'}^{(k')}\right)$$

$$= \epsilon \chi_{c^*} - \epsilon \chi_{c^*} x \left(\sum_{s=0}^{K-1} c_s\right)^{-1} \sum_{k'=0}^{K-1} c_{k'} \cos\left(\theta_{\alpha,\alpha'}^{(k')}\right) - \epsilon(1-x)\chi_{c^*} \frac{1}{N} \sum_{k'=0}^{K-1} c_{k'} \cos\left(\theta_{\alpha,\alpha'}^{(k')}\right).$$

By substituting these facts into Eq. (28), we have

$$[\mathcal{C}(\Sigma^* + E) - \Sigma^*]_{\alpha,\alpha'}$$

$$\approx \epsilon((1-x)\chi_{\kappa} + \chi_{c^*}) - \epsilon \chi_{c^*} x \left(\sum_{s=0}^{K-1} c_s\right)^{-1} \sum_{k'=0}^{K-1} c_{k'} \cos\left(\theta_{\alpha,\alpha'}^{(k')}\right)$$

$$- \epsilon(1-x) \underbrace{\frac{1}{N}(\chi_{\kappa} + \chi_{c^*} - \chi_{q^*})}_{=\chi_{\kappa} x \left(\sum_{s=0}^{K-1} c_s\right)^{-1}} \sum_{k'=0}^{K-1} c_{k'} \cos\left(\theta_{\alpha,\alpha'}^{(k')}\right)$$

$$= ((1-x)\chi_{\kappa} + \chi_{c^*}) \epsilon \underbrace{\left(1 - x \left(\sum_{s=0}^{K-1} c_s\right)^{-1} \sum_{k'=0}^{K-1} c_{k'} \cos\left(\theta_{\alpha,\alpha'}^{(k')}\right)\right)}_{=E_{\alpha,\alpha'}}.$$

Finally, we obtain Eq. (27). □

## B  Proof of Theorem 3.5

**Theorem 3.5** (Trainability). *Let $\tilde{\Sigma}^{(\ell)} \in \mathbb{R}^{N \times N}$ be the gradient covariance with respect to some loss $\mathcal{L}$, e.g. mean squared error, at the $\ell$-th layer. Suppose that the gradient covariance at the $L$-th layer is decomposed as $\tilde{\Sigma}_{\alpha,\alpha'}^{(L)} = \sum_{k=0}^{K-1} \tilde{\epsilon}_k \cos\left(\theta_{\alpha,\alpha'}^{(k)}\right) + \tilde{e}$, where $\tilde{\epsilon}_k$ is the coefficient of each basis and $\tilde{e}$ belongs to the orthogonal complements of $\mathrm{span}(\{\cos\left(\theta_{\alpha,\alpha'}^{(k)}\right)\}_{k=0}^{K-1})$. Then, the gradient covariance*

*at the ℓ-th layer is obtained by*

$$\tilde{\Sigma}^{(\ell)}_{\alpha,\alpha'} = \sum_{k=0}^{K-1} \chi_{c^*}^{L-\ell} \tilde{\epsilon}_k \cos\left(\theta^{(k)}_{\alpha,\alpha'}\right).$$

*Proof.* Recall the definition of the gradient covariance. We first demonstrate the iterated map of the gradient covariance, starting from this definition.

$$\tilde{\Sigma}^{(\ell)}_{\alpha,\alpha'} := \mathbb{E}_{\Theta^{\ell:L}, \Xi^{\ell:L}}\left[g^{(\ell)}_{\alpha,j} g^{(\ell)}_{\alpha',j}\right], \quad g^{(\ell)}_{\alpha,j} := \frac{\partial \mathcal{L}}{\partial H^{(\ell)}_{\alpha,j}} = \sum_{i=1}^{D} \sum_{\beta=0}^{N-1} \frac{\partial \mathcal{L}}{\partial H^{(\ell+1)}_{\beta,i}} \frac{\partial H^{(\ell+1)}_{\beta,i}}{\partial H^{(\ell)}_{\alpha,j}}.$$

The Jacobian $\frac{\partial H^{(\ell+1)}_{\beta,i}}{\partial H^{(\ell)}_{\alpha,j}}$ is calculated as follows.

$$\frac{\partial H^{(\ell+1)}_{\beta,i}}{\partial H^{(\ell)}_{\alpha,j}}$$

$$= \sum_{k=0}^{K-1} \sqrt{\frac{c_k}{2}} \left(F^\dagger_{\beta,k} F_{k,\alpha}\left(\Theta^{(\ell+1,k)}_{j,i} + \sqrt{-1}\Xi^{(\ell+1,k)}_{j,i}\right) + \overline{F^\dagger_{\beta,k} F_{k,\alpha}}\left(\Theta^{(\ell+1,k)}_{j,i} - \sqrt{-1}\Xi^{(\ell+1,k)}_{j,i}\right)\right) \phi'\left(H^{(\ell)}_{\alpha,j}\right)$$

$$= \frac{1}{N} \sum_{k=1}^{K-1} 2\sqrt{\frac{c_k}{2}} \left(\Theta^{(\ell+1,k)}_{j,i} \cos\left(\theta^{(k)}_{\beta,\alpha}\right) + \Xi^{(\ell+1,k)}_{j,i} \sin\left(\theta^{(k)}_{\beta,\alpha}\right)\right) \phi'(H^{(\ell)}_{\alpha,j}).$$

The covariance of Jacobian $\frac{\partial H^{(\ell+1)}_{\beta,i}}{\partial H^{(\ell)}_{\alpha,j}}$ is as follows.

$$\mathbb{E}_{\Theta^{\ell:L}, \Xi^{\ell:L}}\left[\frac{\partial H^{(\ell+1)}_{\beta,i}}{\partial H^{(\ell)}_{\alpha,j}} \frac{\partial H^{(\ell+1)}_{\beta',i}}{\partial H^{(\ell)}_{\alpha',j}}\right]$$

$$= \frac{\sigma^2}{N^2 D} \mathbb{E}_{\mathbf{H}_{:,d} \sim \mathcal{N}(0,\Sigma^{(\ell)})}\left[\phi'(H^{(\ell)}_{\alpha,j})\phi'(H^{(\ell)}_{\alpha,j})\right]$$

$$\times \left(\sum_{k=0}^{K-1} c_k \left(\cos\left(\theta^{(k)}_{\beta,\alpha}\right)\cos\left(\theta^{(k)}_{\beta',\alpha'}\right) + \sin\left(\theta^{(k)}_{\beta,\alpha}\right)\sin\left(\theta^{(k)}_{\beta',\alpha'}\right)\right)\right)$$

$$= \frac{\sigma^2}{N^2 D} \mathbb{E}_{\mathbf{H}_{:,d} \sim \mathcal{N}(0,\Sigma^{(\ell)})}\left[\phi'(H^{(\ell)}_{\alpha,j})\phi'(H^{(\ell)}_{\alpha,j})\right] \left(\sum_{k=0}^{K-1} c_k \cos\left(\theta^{(k)}_{\beta-\beta',\alpha-\alpha'}\right)\right).$$

Then, the covariance of the gradient is given by the following recurrence relation for all $\alpha, \alpha' \in [N]$,

$$\tilde{\Sigma}^{(\ell)}_{\alpha,\alpha'} := \mathbb{E}_{\Theta^{\ell:L}, \Xi^{\ell:L}}\left[g^{(\ell)}_{\alpha,j} g^{(\ell)}_{\alpha',j}\right]$$

$$= \sum_{i,i'=1}^{D} \sum_{\beta,\beta'=0}^{N-1} \mathbb{E}_{\Theta^{\ell:L}, \Xi^{\ell:L}}\left[g^{(\ell+1)}_{\beta,i} g^{(\ell+1)}_{\beta',i'} \frac{\partial H^{(\ell+1)}_{\beta,i}}{\partial H^{(\ell)}_{\alpha,j}} \frac{\partial H^{(\ell+1)}_{\beta',i'}}{\partial H^{(\ell)}_{\alpha',j}}\right]$$

$$= \sum_{\beta,\beta'=0}^{N-1} \sum_{i,i'=1}^{D} \mathbb{E}_{\Theta^{\ell+1:L}, \Xi^{\ell+1:L}}\left[g^{(\ell+1)}_{\beta,i} g^{(\ell+1)}_{\beta',i'}\right] \mathbb{E}_{\Theta^{\ell:L}, \Xi^{\ell:L}}\left[\frac{\partial H^{(\ell+1)}_{\beta,i}}{\partial H^{(\ell)}_{\alpha,j}} \frac{\partial H^{(\ell+1)}_{\beta',i'}}{\partial H^{(\ell)}_{\alpha',j}}\right] \delta_{i,i'}$$

$$= \frac{\sigma^2}{N^2} \sum_{\beta,\beta'=0}^{N-1} \tilde{\Sigma}^{(\ell+1)}_{\beta,\beta'} \mathbb{E}_{\mathbf{H}_{:,d} \sim \mathcal{N}(0,\Sigma^{(\ell)})}\left[\phi'(H^{(\ell)}_{\alpha,j})\phi'(H^{(\ell)}_{\alpha',j})\right] \left(\sum_{k=0}^{K-1} c_k \cos\left(\theta^{(k)}_{\beta-\beta',\alpha-\alpha'}\right)\right).$$

As with (Schoenholz et al., 2016), we approximate $\Sigma^{(\ell+1)} \approx \Sigma^*$ since the number of layer $\ell$ is assumed to be sufficiently large. Then, the linear iterated map of the gradient covariance $\Sigma^{(\ell+1)} \mapsto \Sigma^{(\ell)}$ is given as follows.

$$\tilde{\Sigma}^{(\ell)}_{\alpha,\alpha'} = \frac{1}{N^2} \sum_{\beta,\beta'=0}^{N-1} \tilde{\Sigma}^{(\ell+1)}_{\beta,\beta'} \chi_{c^*} \left(\sum_{k=0}^{K-1} c_k \cos\left(\theta^{(k)}_{\beta-\beta',\alpha-\alpha'}\right)\right). \tag{29}$$

The rank of the linear iterated map $\boldsymbol{\Sigma}^{(\ell+1)} \mapsto \boldsymbol{\Sigma}^{(\ell)}$ is less than $K$ since the matrix representation of the linear map can be decomposed into two matrices $\tilde{\boldsymbol{A}} \in \mathbb{R}^{N^2 \times K}$ and $\tilde{\boldsymbol{B}} \in \mathbb{R}^{K \times N^2}$ as follows.

$$\tilde{\Sigma}_{\alpha,\alpha'}^{(\ell)} = \frac{1}{N^2} \sum_{k=0}^{K-1} c_k \exp\left(-\sqrt{-1}\theta_{\beta,\beta'}^{(k)}\right) \exp\left(\sqrt{-1}\theta_{\alpha',\alpha}^{(k)}\right) \chi_{c^*} \tilde{\Sigma}_{\beta,\beta'}^{(\ell+1)} = \left(\sum_{k=0}^{K-1} \tilde{A}_{(\alpha,\alpha'),k} \tilde{B}_{k,(\beta,\beta')}\right) \tilde{\Sigma}_{\beta,\beta'}^{(\ell+1)},$$

$$\tilde{A}_{(\alpha,\alpha'),k} := \frac{1}{N^2} c_k \exp\left(\sqrt{-1}\theta_{\alpha',\alpha}^{(k)}\right) \chi_{c^*}, \quad \tilde{B}_{k,(\beta,\beta')} := \exp\left(-\sqrt{-1}\theta_{\beta,\beta'}^{(k)}\right).$$

Next, we show that the subspace $\mathrm{span}\left(\{\cos\left(\theta_{\alpha,\alpha'}^{(k)}\right)\}_{k=0}^{K-1}\right) \subset \mathbb{R}^{N \times N}$ is the $K$-dimensional eigenspace with eigenvalue $\chi_{c^*}$ of the linear iterated map $\boldsymbol{\Sigma}^{(\ell+1)} \mapsto \boldsymbol{\Sigma}^{(\ell)}$.

By substituting $\tilde{\Sigma}_{\beta,\beta'}^{(\ell+1)} = \sum_{k=0}^{K-1} \chi_{c^*}^{L-(\ell+1)} \tilde{\epsilon}_k \cos\left(\theta_{\beta,\beta'}^{(k)}\right)$ into Eq. (29), we obtain

$$\tilde{\Sigma}_{\alpha,\alpha'}^{(\ell)} = \frac{1}{N^2} \sum_{\beta,\beta'=0}^{N-1} \chi_{c^*} \left(\sum_{k=0}^{K-1} \tilde{\epsilon}_k \cos\left(\theta_{\beta,\beta'}^{(k)}\right)\right) \sum_{k'=0}^{K-1} c_{k'} \cos\left(\theta_{\beta-\beta',\alpha-\alpha'}^{(k')}\right).$$

From the orthogonality of the cosine and sine function, we obtain

$$\frac{1}{N^2} \sum_{\beta,\beta'=0}^{N-1} \sum_{k'=0}^{K-1} c_{k'} \cos\left(\theta_{\beta-\beta',\alpha-\alpha'}^{(k')}\right) \cos\left(\theta_{\beta,\beta'}^{(k)}\right)$$

$$= \frac{1}{N^2} \sum_{\beta,\beta'=0}^{N-1} \sum_{k'=0}^{K-1} c_{k'} \left(\cos\left(\theta_{\beta,\beta'}^{(k')}\right) \cos\left(\theta_{\alpha',\alpha}^{(k')}\right) \cos\left(\theta_{\beta,\beta'}^{(k)}\right)\right.$$

$$\left. + \sin\left(\theta_{\beta,\beta'}^{(k')}\right) \sin\left(\theta_{\alpha',\alpha}^{(k')}\right) \cos\left(\theta_{\beta,\beta'}^{(k)}\right)\right) \tag{30}$$

$$= \cos\left(\theta_{\alpha,\alpha'}^{(k)}\right).$$

Hence, we have

$$\tilde{\Sigma}_{\alpha,\alpha'}^{(\ell)} = \sum_{k=0}^{K-1} \chi_{c^*}^{L-\ell} \tilde{\epsilon}_k \cos\left(\theta_{\alpha,\alpha'}^{(k)}\right).$$

This completes the proof. $\qquad\square$

## C Discussions on the similarity of DCN and CNN

While CNNs perform local convolutions in the spatial domain, FNOs execute convolutions in the frequency domain, thereby achieving global convolutions in the spatial domain. Our theory for the FNO and the mean-field theory for CNNs (Xiao et al., 2018) share a common focus on the correlation dynamics $\boldsymbol{\Sigma}^{(0)}, \boldsymbol{\Sigma}^{(1)}, \dots, \boldsymbol{\Sigma}^{(L)}$ of the spatial representations $\mathbf{H}^{(0)}, \mathbf{H}^{(1)}, \dots, \mathbf{H}^{(L)} \in \mathbb{R}^{N \times D}$. The iterated maps for CNNs and FNOs are obtained as follows.

$$\Sigma_{\alpha,\alpha'}^{(\ell+1)} = \frac{\sigma^2}{2r+1} \sum_{\beta \in \ker} \mathbb{E}_{\mathbf{H}_{:,d} \sim \mathcal{N}(\mathbf{0}, \boldsymbol{\Sigma}^{(\ell)})} \left[\phi(H_{\alpha+\beta,d})\phi(H_{\alpha'+\beta,d})\right] + \sigma_b^2 =: \mathcal{C}_{\mathrm{CNN}}(\boldsymbol{\Sigma}^{(\ell)}),$$

$$\Sigma_{\alpha,\alpha'}^{(\ell)} = \sigma^2 \sum_{k=0}^{K-1} c_k \mathbb{E}_{\mathbf{H}_{:,d} \sim \mathcal{N}(\mathbf{0}, \boldsymbol{\Sigma}^{(\ell)})} \left[\left|[\boldsymbol{F}\phi(\mathbf{H}_{:,d})]_k\right|^2\right] \cos\left(\theta_{\alpha,\alpha'}^{(k)}\right) + \sigma_b^2 =: \mathcal{C}_{\mathrm{FNO}}(\boldsymbol{\Sigma}^{(\ell)}),$$

where $2r+1$ is the number of filter width and $\ker = \{\beta \in \mathbb{Z} | |\beta| \leq r\}$ is set of indices referring to the elements of the filter.

When we consider the FNO without mode truncation ($K = N/2 + 1$), the propagation of the diagonal components $(\alpha, \alpha)$ for any $\alpha \in [N]$ is equivalent to a CNN with filter size $N$ performing global

convolution.

$$\Sigma_{\alpha,\alpha}^{(\ell)} = \sigma^2 \sum_{k=0}^{\frac{N}{2}} c_k \mathbb{E}_{\mathbf{H}_{:,d}\sim\mathcal{N}(\mathbf{0},\boldsymbol{\Sigma}^{(\ell)})} \left[\left|[\boldsymbol{F}\phi(\mathbf{H}_{:,d})]_k\right|^2\right] \underbrace{\cos\left(\theta_{\alpha,\alpha}^{(k)}\right)}_{=1} + \sigma_b^2$$

$$= \frac{\sigma^2}{N} \sum_{\beta=0}^{N-1} \mathbb{E}_{\mathbf{H}_{:,d}\sim\mathcal{N}(\mathbf{0},\boldsymbol{\Sigma}^{(\ell)})} \left[|\phi(H_{\beta,d})|^2\right] + \sigma_b^2$$

$$= \frac{\sigma^2}{N} \sum_{\beta=0}^{N-1} \mathbb{E}_{\mathbf{H}_{:,d}\sim\mathcal{N}(\mathbf{0},\boldsymbol{\Sigma}^{(\ell)})} \left[\phi(H_{\alpha+\beta,d})\phi(H_{\alpha+\beta,d})\right] + \sigma_b^2.$$

The first equality follows from Perseval's equality and the second from the periodic boundary condition. In contrast, the propagation of the off-diagonal components differs between CNN and FNO, and the iterated map is different even in the presence of mode truncation.

Regarding the fixed point, Xiao et al. (2018) demonstrated that any fixed point for the iterated map of the DCN is also a fixed point for that of the CNN. Consequently, Lemma A.1 indicates that the fixed points for CNN and FNO are consistent.

The behaviour of the iterated map around fixed points reflects the nature of each architecture. CNN possess diagonal eigenspaces associated with eigenvalues $\chi_{q^*}$ and non-diagonal eigenspaces associated with eigenvalues $\chi_{c^*}$. FNOs without mode truncation exhibit a similarity, possessing eigenspaces $\chi_{q^*}$ for zero-frequency and eigenspaces $\chi_{c^*}$ for k-frequencies with diagonal components removed.

Finally, the iterated map during the backpropagation for CNN and FNO are given by

$$\tilde{\Sigma}_{\alpha,\alpha'}^{(\ell)} = \sigma^2 \sum_{\beta\in\ker} v_\beta \tilde{\Sigma}_{\alpha-\beta,\alpha'-\beta}^{(\ell+1)} \mathbb{E}_{\mathbf{H}_{:,d}\sim\mathcal{N}(0,\boldsymbol{\Sigma}^{(\ell)})} \left[\phi'(H_{\alpha,j}^{(\ell)})\phi'(H_{\alpha',j}^{(\ell)})\right]$$

$$\tilde{\Sigma}_{\alpha,\alpha'}^{(\ell)} = \frac{\sigma^2}{N^2} \sum_{\beta,\beta'=0}^{N-1} \tilde{\Sigma}_{\beta,\beta'}^{(\ell+1)} \mathbb{E}_{\mathbf{H}_{:,d}\sim\mathcal{N}(0,\boldsymbol{\Sigma}^{(\ell)})} \left[\phi'(H_{\alpha,j}^{(\ell)})\phi'(H_{\alpha',j}^{(\ell)})\right] \left(\sum_{k=0}^{K-1} c_k \cos\left(\theta_{\beta-\beta',\alpha-\alpha'}^{(k)}\right)\right),$$

where $v_\beta$ is the variance weight parameter dependent of the filter position $\beta$, i.e., $w_{i,j}(\beta) \sim \mathcal{N}(0, \sigma^2 v_\beta/D)$, $\sum_{\beta\in\ker} v_\beta = 1$.

Using the approximation $\boldsymbol{\Sigma}^{(\ell+1)} \approx \boldsymbol{\Sigma}^*$ as with (Schoenholz et al., 2016), the backpropagation of the diagonal components $(\alpha,\alpha)$ of the FNO without mode truncation is equivalent to that of the global CNN (see Eq. (2.16) in (Xiao et al., 2018)) with $v_\beta = \frac{1}{N}$, as shown below:

$$\tilde{\Sigma}_{\alpha,\alpha}^{(\ell)} \approx \frac{1}{N^2} \sum_{\beta,\beta'=0}^{N-1} \tilde{\Sigma}_{\beta,\beta'}^{(\ell+1)} \chi_{c^*} \underbrace{\left(\sum_{k=0}^{N/2} c_k \cos\left(\theta_{\beta-\beta',\alpha-\alpha}^{(k)}\right)\right)}_{=N\delta_{\beta,\beta'}} = \chi_{c^*} \sum_{\beta=0}^{N-1} \frac{1}{N} \tilde{\Sigma}_{\beta,\beta}^{(\ell+1)}.$$

This equivalence suggests that the edge of chaos initialization (e.g., He initialization) is also valid for the FNO since the problems of gradient vanishing and explosion are determined by the diagonal components of $\tilde{\boldsymbol{\Sigma}}$.

# D   Details of Experimental Setup

In this section, we summarize the detailed setup of the all experiments, including the experiments in Section 4.

## D.1   Datasets

### D.1.1   Advection equation

We used the advection equation data published by Takamoto et al. (2022). The advection equation for the function $u(x,t) \in L^2((0,1) \times (0,2]; \mathbb{R})$ is given by

$$\partial_t u(x,t) + \beta \partial_x (u(x,t)/2) = 0, \ u(x,0) = u_0(x),$$

where $u_0 \in L^2((0,1); \mathbb{R})$ is the initial condition and $\beta \in \mathbb{R}$ is an advection speed set to 2.0. The exact solution is given as $u(x, t) = u_0(x - \beta t)$ for any initial condition $u_0$.

Only periodic boundary conditions were used in this dataset. The initial conditions are the superposition of sinusoidal wave given by

$$u_0(x) = \sum_{i=1}^{k_{\max}} A_i \sin(k_i x + \phi_i), \tag{31}$$

where $k_i = 2\pi \sum_{j=1}^N n_{i,j}/L_x$ are wave numbers whose $n_{i,j}$ are integer numbers randomly chosen in $[1, k_{\max}]$, $L_x = 1$ is the calculation domain size, $N = 2$ is the number of wave to be added, and $k_{\max} = 8$ is the maximum wave number. The amplitude $A_i$ is uniformly chosen in $[0, 1]$, and the phase $\phi_i$ is the randomly chosen in $(0, 2\pi)$. The 2nd-order temporal and spatial upwind finite difference scheme was used for generating the data. Settings are described in Appendix D of (Takamoto et al., 2022).

### D.1.2 Burgers' equation

We used the Burgers' equation data published by Takamoto et al. (2022). The Burgers' equation for the function $u(x, t) \in L^2((0, 1) \times (0, 2]; \mathbb{R})$ is given by

$$\partial_t u(x, t) + \partial_x \left( u^2(x, t)/2 \right) = \nu \partial_{xx} u(x, t),$$
$$u(x, 0) = u_0(x),$$

where $u_0 \in L^2((0, 1); \mathbb{R})$ is the initial condition and $\nu$ is the diffusion coefficient set to 4.0. The periodic boundary conditions and Equation (31) are used as the initial conditions. The 2nd-order temporal and spatial upwind finite difference scheme is used for generating the data. Settings are described in Appendix D of (Takamoto et al., 2022).

### D.1.3 Darcy Flow equation

We used the data of 2D Darcy Flow equation on a regular grid published by Li et al. (2020a). The Darcy Flow equation for the function $u \in H_0^1((0, 1)^2; \mathbb{R}_+)$ with a Dirichlet boundary is given by

$$-\nabla \cdot (a(x)\nabla u(x)) = f(x), \qquad\qquad x \in (0, 1)^2,$$
$$u(x) = 0, \qquad\qquad x \in \partial(0, 1)^2,$$

where $a \in L^\infty((0, 1)^2; \mathbb{R}_+)$ is the diffusion coefficient and $f \in L^2((0, 1)^2; \mathbb{R})$ is the forcing function. The coefficients $a$ was generated by measure $\mu = \psi_\sharp \mathcal{N}(0, (-\Delta + 9I)^{-2})$ using the Laplacian with zero Neumann boundary and the binary point-wise mapping $\psi(x) = 12 \ (x \geq 0), \ 3 \ (x < 0)$. The forcing function is fixed $f(x) = 1$. Our aim is to predict the operator mapping the diffusion coefficient to the solution $a \to u$. The solution function $u$ was generated by using the second-order finite difference scheme on a $421 \times 421$ grid. Settings are described in Appendix A.3.2 of (Li et al., 2020c).

### D.1.4 Incompressible Navier-Stokes equation

We used the 2D NS equation on the unit torus defined by

$$\partial_t \omega(x, t) + u(x, t) \cdot \nabla \omega(x, t) = \nu \nabla^2 \omega(x, t) + f(x),$$
$$\nabla \cdot u(x, t) = 0, \ \omega(x, 0) = \omega_0(x),$$

where $\omega(x, t) \in C([0, T]; H^r((0, 1)^2; \mathbb{R}^2))$ is the vorticity, $\omega_0 \in H^r((0, 1)^2; \mathbb{R}^2)$ is the initial vorticity, $u(x, t) \in C([0, T]; H^r((0, 1)^2; \mathbb{R}^2))$ is the velocity field for any $r > 0$, $\nu \in \mathbb{R}_+$ is the viscosity, and $f \in L^2((0, 1)^2; \mathbb{R})$ is the external forcing function defined by $f(x) = 0.1 \left( \sin(2\pi(x_1 + x_2)) + \cos(2\pi(x_1 + x_2)) \right)$. The initial vorticity $\omega_0$ was generated by $\omega_0 \sim \mu$ where $\mu = \mathcal{N}(0, 7^{\frac{3}{2}}(-\Delta + 49I)^{-2.5})$ with periodic boundary conditions. The viscosity was set to $1e{-}3$, $1e{-}4$, or $1e{-}5$. Our aim is to predict the operator that maps a solution $u$ up to time 10 to a solution up to some later time $T > 10$. The data was generated by the pseudo-spectral Crak-Nicholson second-order method on $64 \times 64$ grid. For the data with $\nu = 1e{-}4$, the time resolution was also downsampled by half. Settings are described in Section 5.3 of (Li et al., 2020c).

Table 2: Training settings

| PDE | architecture | batch size | initial lr | max_epochs |
|---|---|---|---|---|
| advection | Simplified FNO with Tanh | 40 | $5.0 \times 10^{-5}$ | 200 |
| advection | Simplified FNO with ReLU | 40 | $5.0 \times 10^{-5}$ | 200 |
| Burgers' | Simplified FNO with Tanh | 40 | $5.0 \times 10^{-5}$ | 200 |
| Burgers' | Simplified FNO with ReLU | 40 | $1.0 \times 10^{-3}$ | 200 |
| Darcy flow | 2D FNO with ReLU | 20 | $1.0 \times 10^{-3}$ | 500 |
| Navier-Stokes ($\nu = 1e$-3) | 2D FNO with ReLU | 20 | $1.0 \times 10^{-4}$ | 500 |
| Navier-Stokes ($\nu = 1e$-4) | 2D FNO with ReLU | 50 | $2.5 \times 10^{-3}$ | 400 |
| Navier-Stokes ($\nu = 1e$-5) | 2D FNO with ReLU | 20 | $2.5 \times 10^{-3}$ | 500 |

### D.2 Training settings

Our detailed training settings of the experiments in Section 4 are provided in Table 2. Our experimental environment consists of an Intel Xeon Plantinum 8360Y (36-core) CPU and a single NVIDIA A100 GPU. Most of our code for experiments are based on the code of PDEBench (`https://github.com/pdebench/PDEBench`) (Takamoto et al., 2022). The only modifications to the model are to multiply the outputs (variable `out_ft` in code of class FNO1d and FNO2d) corresponding to mode $k = 0, N/2$ by $\sqrt{2}$ and to initialize the weights by Gaussian distribution, as described in Section 3.1.

## E  Detailed Experimental Analysis

### E.1  Analysis of Training Loss

Figure 6a shows the training loss for each epoch of the 32-layer FNOs with parameters $\sigma^2 \in \{0.1, 0.5, 1.0, 2.0, 3.0, 4.0\}$ on the NS dataset with $\nu = 1e-3$. When the initial parameter $\sigma^2$ is too small, the training loss is not well reduced due to gradient vanishing. On the other hand, when the initial parameter $\sigma^2$ is too large, the initial training loss blows up due to gradient exploding. The proposed edge of chaos initialization smoothly reduces the training loss in the initial epoch and enables stable training.

### E.2  Analysis of Test Performance

The nMSE of the FNOs on test datasets for six distinct PDEs is presented in Tables 3 and 4. Results are shown only for the FNOs with initial parameters where training was successful in many cases. For the NS equation with viscosity values of $\nu = 1e-3, 1e-4$, where sufficient data is available, Table 4 shows that best performance is achieved with 8 or 16 layers. This suggests that while shallow FNOs are currently prevalent, deep FNOs could be advantageous in certain tasks, underscoring the significance of our analysis of the bias in deep FNOs. Conversely, for other equations, the 4-layer FNO performs best and deeper FNOs result in a drop in performance even with the edge of chaos initialization. We will discuss this test performance deterioration in detail.

The over-fitting phenomenon is observed in the Darcy Flow and NS equation datasets with $\nu = 1e-5$, where only limited training data is available. The training loss for each epoch on the NS equation is depicted in Fig. 6b. As demonstrated in Fig. 6b, the 16 and 32-layer FNOs yield a lower training loss than the 4-layer FNO, but exhibit poorer performance on the test dataset as shown in Table 4. These results suggest over-fitting to the training data, necessitating either abundant training data or appropriate regularization.

Conversely, the under-fitting phenomenon is apparent in the 1D advection and Burgers' equation datasets in Table 3. The training loss of the FNO with ReLU activation for each epoch on the Burgers' equation is presented in Fig. 6c. Figure 6c indicates that the larger the number of layers, the higher the training loss in the final epoch, and the worse the test performance. This under-fitting to the training data could be attributed to the escalating complexity of the loss landscape as the layer count increases, a known issue for DCN and CNN (Li et al., 2018). This may be due to the emergence of local minima corresponding to operators that generate too complex functions, preventing the

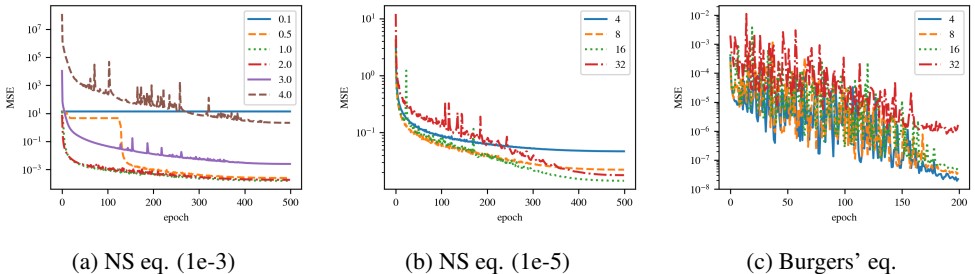

|  | (a) NS eq. (1e-3) | (b) NS eq. (1e-5) | (c) Burgers' eq. |

Figure 6: Training Loss Curve. (a): training loss curve of the 32-layer original FNOs with varying initial parameters $\sigma^2 \in \{0.1, 0.5, 1.0, 2.0, 3.0, 4.0\}$, on the NS equation with $\nu = 1e-3$. (b): training loss curve of the original FNOs with an initial parameter $\sigma^2 = 2.0$ with a varying number of layers $L \in \{4, 8, 16, 32\}$ on the NS equation with $\nu = 1e-5$. (c): training loss curve of the simplified FNOs with ReLU activation and the initial parameter $\sigma^2 = 2.0$ with varying number of layers $L \in \{4, 8, 16, 32\}$ on the Burgers' equation.

Table 3: Test performance measured by nMSE of 1D simplified FNO on 1D PDEs

| nMSE | Layers | Advection | | Burgers' | |
| | | Tanh | ReLU | Tanh | ReLU |
|---|---|---|---|---|---|
| $\sigma^2 = 1.0$ | 4 | 0.013 | 0.015 | 0.0055 | 0.00088 |
| | 8 | 0.013 | 0.015 | 0.0069 | 0.0012 |
| | 16 | 0.013 | 0.015 | 0.0068 | 0.0016 |
| | 32 | 0.016 | 0.018 | 0.0071 | 0.0041 |
| $\sigma^2 = 2.0$ | 4 | 0.013 | 0.017 | 0.0036 | 0.00098 |
| | 8 | 0.012 | 0.018 | 0.0034 | 0.0011 |
| | 16 | 0.012 | 0.020 | 0.0050 | 0.0016 |
| | 32 | 0.014 | 0.024 | 0.0062 | 0.0027 |
| $\sigma^2 = 3.0$ | 4 | 0.013 | 0.019 | 0.0044 | 0.00096 |
| | 8 | 0.014 | 0.022 | 0.0042 | 0.0012 |
| | 16 | 0.020 | 0.032 | 0.0060 | 0.0016 |
| | 32 | 0.053 | 0.059 | 0.0093 | 0.0045 |

attainment of parameters that achieve global minima. This issue could be mitigated by introducing more suitable regularization, an appropriate optimizer, or a skip connection (Tran et al., 2022).

Our theory and experiments suggest that the training of deep FNOs has suffered from problems including gradient vanishing and exploding due to improper initialization, over-fitting caused by insufficient training data, and under-fitting caused by loss landscapes with strong non-convexity. While our edge of chaos initialization prevents the gradient vanishing and exploding, techniques to solve over-fitting and under-fitting problems are still needed in practice.

## F   Visualization of Forward Propagation

We visualized the behavior of the simplified FNO's covariance matrix $\Sigma^{(\ell)}$ with varying initialization parameters $\sigma^2 \in \{0.1, 1.0, 2.0, 4.0\}$ and activation functions. The FNO, with a width of $D = 1024$, was used and the input was sampled from the standard normal distribution with a spatial size of $N = 32$. The results of the FNO with Tanh activation, both with and without mode truncation, are shown in Figs. 7 and 8 and Figs. 9 and 10 respectively. Similarly, the results of the FNO with ReLU activation, both with and without mode truncation, are displayed in Figs. 11 and 12 and Figs. 13 and 14 respectively. In the ordered phase, all figures illustrate convergence to the fixed point $\Sigma^*$ where $c^* = 1$, with the rate of convergence increasing as the parameter $\sigma^2$ decreases. In the chaotic phase, the activation function dictates the covariance behavior. Without mode truncation, the covariance behavior of the FNO is identical to those of the DCN; otherwise non-uniform, FNO-specific periodic covariance is exhibited.

Table 4: Test performance measured by nMSE of 2D original FNO with ReLU activation on Darcy Flow and NS equation.

| nMSE | Layers | Darcy Flow | Navier-Stokes | | |
| --- | --- | --- | --- | --- | --- |
| | | | $\nu = 1e-3$ | $\nu = 1e-4$ | $\nu = 1e-5$ |
| | 4 | 0.025 | 0.0063 | 0.18 | 0.10 |
| $\sigma^2 = 1.0$ | 8 | 0.028 | 0.0047 | 0.14 | 0.094 |
| | 16 | 0.035 | 0.0048 | 0.12 | 0.12 |
| | 32 | 0.56 | 0.0057 | 0.13 | 0.16 |
| | 4 | 0.029 | 0.0075 | 0.18 | 0.10 |
| $\sigma^2 = 2.0$ | 8 | 0.036 | 0.0057 | 0.14 | 0.11 |
| | 16 | 0.041 | 0.0057 | 0.12 | 0.11 |
| | 32 | 0.042 | 0.0072 | 0.13 | 0.18 |
| | 4 | 0.033 | 0.0089 | 0.18 | 0.10 |
| $\sigma^2 = 3.0$ | 8 | 0.040 | 0.0080 | 0.14 | 0.11 |
| | 16 | 0.052 | 0.0098 | 0.13 | 0.13 |
| | 32 | 0.16 | 0.028 | 0.14 | 0.19 |

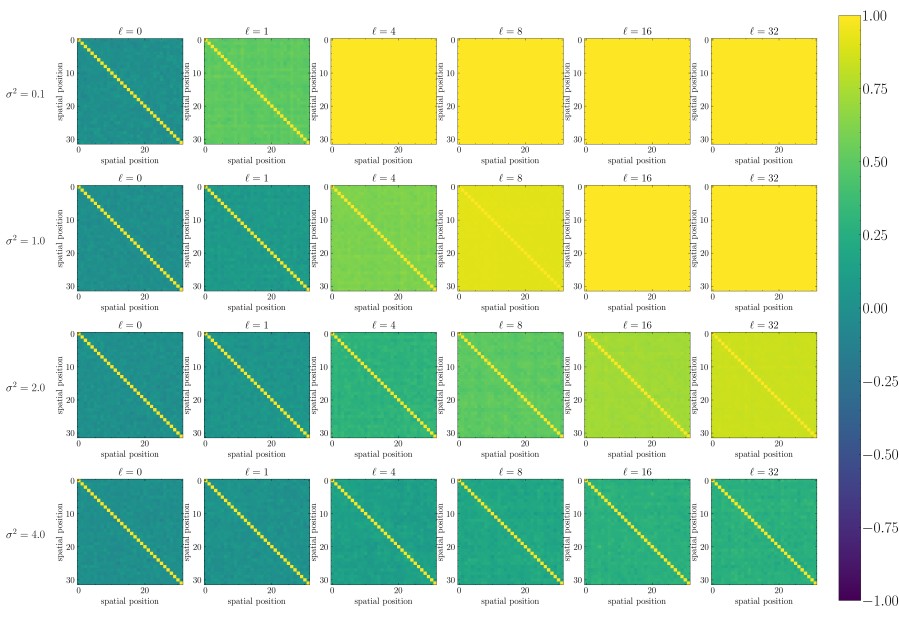

Figure 7: Visualization of the correlation $\Sigma^{(\ell)}_{\beta,\beta'}/\sqrt{\Sigma^{(\ell)}_{\beta,\beta}\Sigma^{(\ell)}_{\beta',\beta'}}$ for the simplified FNO with Tanh activation and no mode truncation.

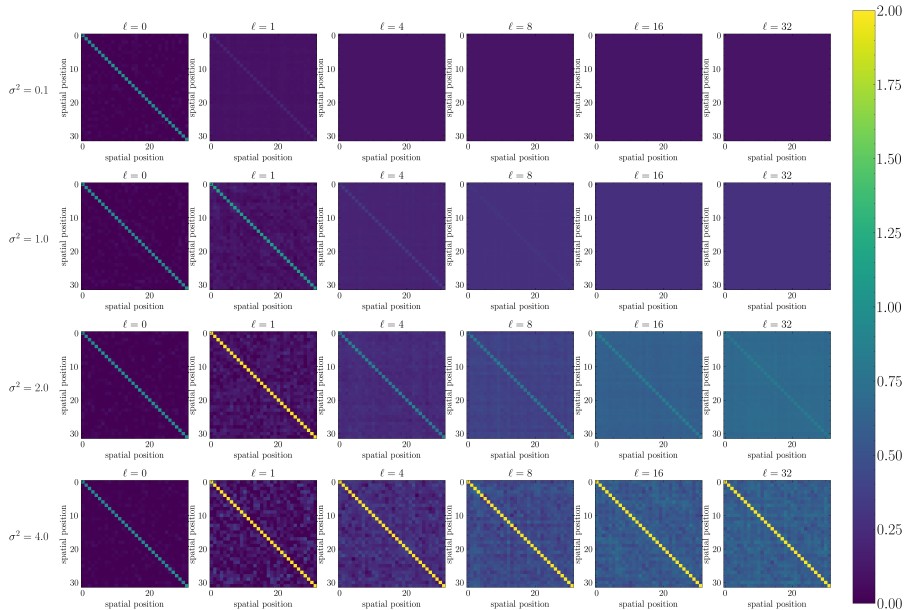

Figure 8: Visualization of the covariance $\mathbf{\Sigma}^{(\ell)}$ for the simplified FNO with Tanh activation and no mode truncation.

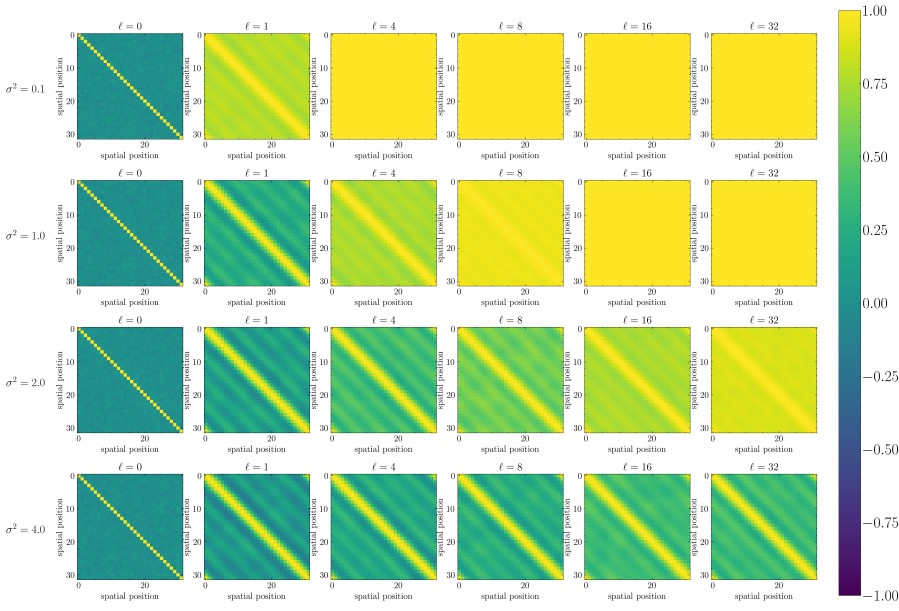

Figure 9: Visualization of the correlation $\Sigma^{(\ell)}_{\beta,\beta'}/\sqrt{\Sigma^{(\ell)}_{\beta,\beta}\Sigma^{(\ell)}_{\beta',\beta'}}$ for the simplified FNO with Tanh activation and the Fourier mode $K = 5$.

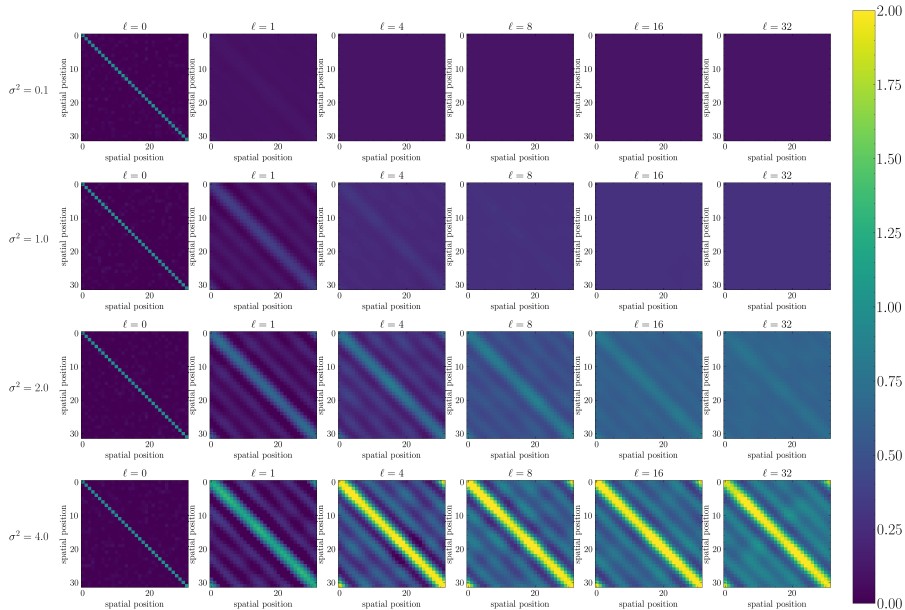

Figure 10: Visualization of the covariance $\boldsymbol{\Sigma}^{(\ell)}$ for the simplified FNO with Tanh activation and the Fourier mode $K = 5$.

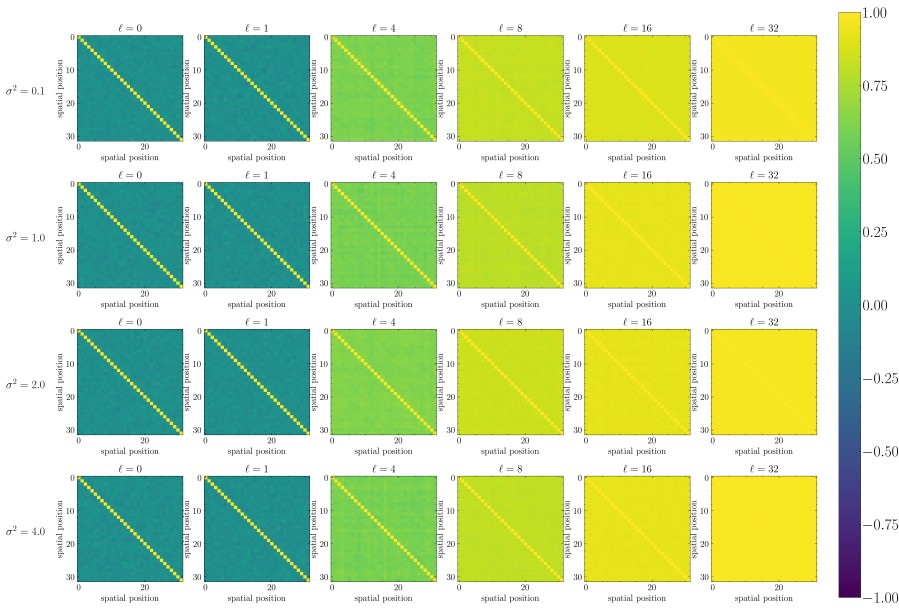

Figure 11: Visualization of the correlation $\Sigma_{\beta,\beta'}^{(\ell)}/\sqrt{\Sigma_{\beta,\beta}^{(\ell)}\Sigma_{\beta',\beta'}^{(\ell)}}$ for the simplified FNO with ReLU activation and no mode truncation.

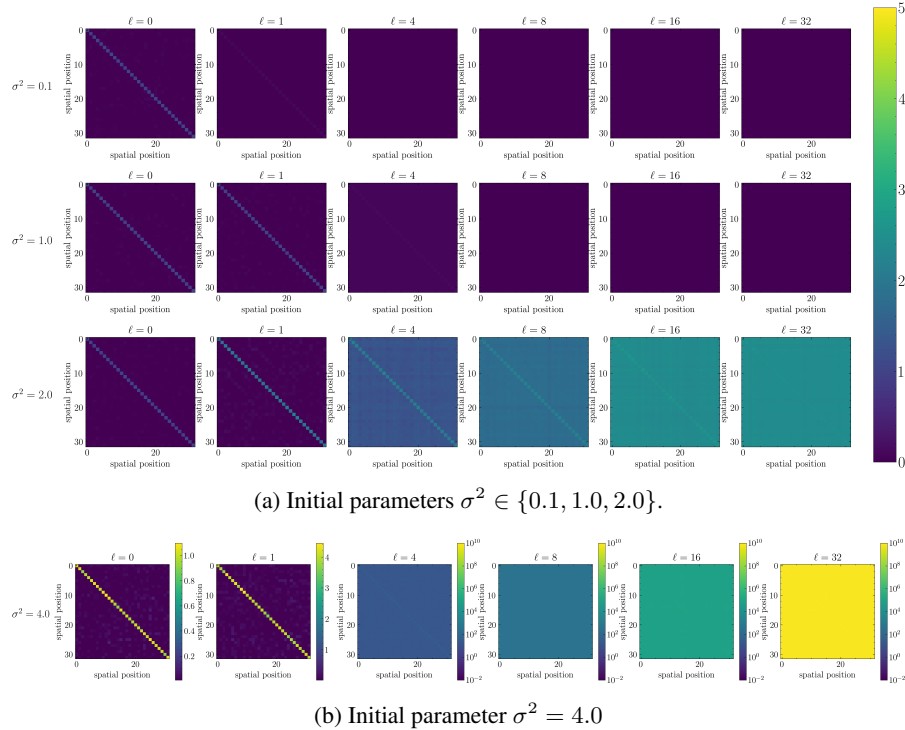

(a) Initial parameters $\sigma^2 \in \{0.1, 1.0, 2.0\}$.

(b) Initial parameter $\sigma^2 = 4.0$

Figure 12: Visualization of the covariance $\mathbf{\Sigma}^{(\ell)}$ for the simplified FNO with ReLU activation and no mode truncation.

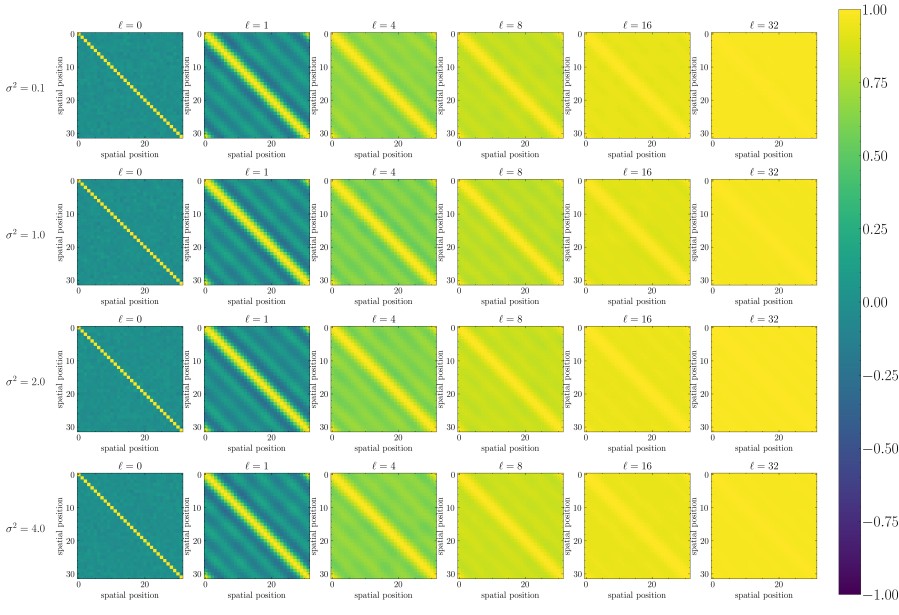

Figure 13: Visualization of the correlation $\Sigma^{(\ell)}_{\beta,\beta'}/\sqrt{\Sigma^{(\ell)}_{\beta,\beta}\Sigma^{(\ell)}_{\beta',\beta'}}$ for the simplified FNO with ReLU activation and the Fourier mode $K = 5$.

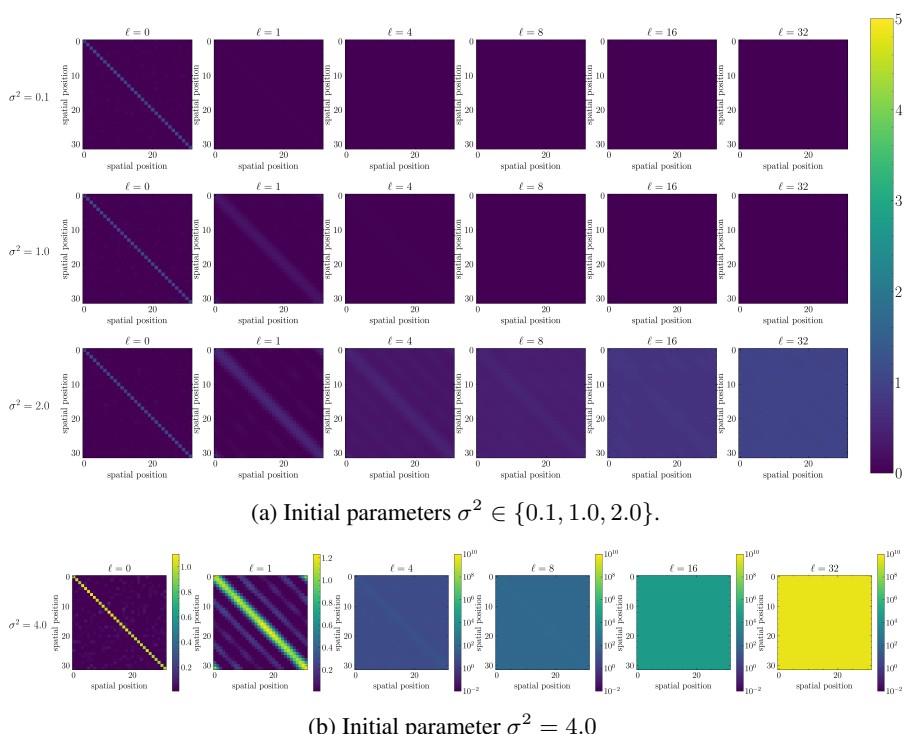

(a) Initial parameters $\sigma^2 \in \{0.1, 1.0, 2.0\}$.

(b) Initial parameter $\sigma^2 = 4.0$

Figure 14: Visualization of the covariance $\boldsymbol{\Sigma}^{(\ell)}$ for the simplified FNO with ReLU activation and the Fourier mode $K = 5$.

