# OpenReview forum: "Understanding the Expressivity and Trainability of Fourier Neural Operator: A Mean-Field Perspective"
_NeurIPS.cc/2024/Conference — NeurIPS 2024 poster_

### Official Review · Reviewer_gSyo · 2024-07-06

**Soundness:** 3
**Presentation:** 3
**Contribution:** 3
**Rating:** 6
**Confidence:** 3

**Summary:**

In this work, the authors investigate the expressive power and trainability of the Fourier Neural Operator (FNO). A mean-field theory is adapted for the FNO to examine the behavior of randomly initialized FNOs from the perspective of the 'edge of chaos'. The investigation focuses on understanding the expressive behavior of randomly initialized FNOs by examining the transition between ordered and chaotic phases. This phase transition demonstrates specific characteristics unique to the FNO, influenced by mode truncation, and shares similarities observed in densely connected networks and CNNs. Additionally, the authors establish a correlation between expressive power and trainability: the ordered and chaotic phases correspond to regions where gradients vanish or explode, respectively. The authors identify the necessity of initializing FNO near the edge of chaos for stable training theoretically and experimentally.

**Strengths:**

1. **Theoretical novelty:** The paper adapts mean-field theory specifically for Fourier Neural Operators (FNOs), extending existing theories from other neural network architectures such as densely connected networks and CNNs.
2. **Practical implications:** The theoretical findings lead to concrete recommendations for initializing FNOs to ensure stable training, particularly for deep FNOs.
3. **Comprehensive experiments:** The authors validate their theoretical results across multiple PDE-solving tasks, including advection, Burgers, Darcy Flow, and the Navier-Stokes equations. All these datasets are publicly available.

**Weaknesses:**

1. The adaptation of mean-field theory to FNOs, while a notable endeavor, may not sufficiently break new ground in the field.
2. The significance of the theoretical aspects that FNOs share similarities with CNNs should be further discussed.
Overall, this work addresses how the network should be initialized for stable training of FNOs; however, its contribution might be somewhat limited. A stability analysis beyond the initialization phase, such as on the architecture, would be interesting to see.
4. There are barely any experimental results shown in the main text, and the discussion of how their theoretical analysis is useful in practical applications is lacking. The discovery in their work, the He distribution for FNOs, has already been applied in prior works such as [1] (see their GitHub implementation, `reset_parameters()` method under models); hwoever, it is always good to have a rigorous justification, which is shown in this work.

[1] Helwig, J., Zhang, X., Fu, C., Kurtin, J., Wojtowytsch, S., and Ji, S. Group equivariant Fourier neural operators for partial differential equations.

**Questions:**

**Major Questions:**

1. If we treat FNO simply as a CNN, where global convolution occurs in the frequency domain, with Parseval theorems, it also implies, though less rigorously, that we can initialize the (complex) weights based on the He Distribution. This also hints on the similarities between FNOs and CNNs. Could the authors comment on this perspective?

2. In line 122, it states:
   > "two learnable weights $\Theta^{(\ell, k)} \in \mathbb{R}^{D \times D}$ and $\Xi^{(\ell, k)} \in \mathbb{R}^{D \times D}$". If I understand correctly, these are complex weights for spectral convolution.

   Firstly, why is $\Xi^{(\ell, k)} \in \mathbb{R}^{D \times D}$? Is this a typo where a $K$ is missing? Secondly, why are there two sets of learnable weights? Isn't there just one for the Fourier transform of the convolution kernel? Do you treat the real and complex parts separately, resulting in one set of parameters for the real and another for the complex parts?

3. The weights in spectral convolution are complex. By initializing the weights according to $\mathcal{N}\left(0, \frac{\sigma^2}{4 D}\right)$, do you treat the real and complex parts separately?

4. The FNO is implemented with $L = 64$ layers, which is particularly deep for FNOs involving $64$ forward and backward FFTs. Is such a large number of layers practical in real-world applications? If not, how are your findings useful for shallower FNOs?

**Minor/Optional Questions and Suggestions:**

1. Some abbreviations are introduced before their full names, such as DCN in line 25, whereas its full name is introduced in line 37 (densely connected network, DCN).

2. Given your insights into FNO expressivity and trainability, do you have any thoughts on potential architectural improvements or variations that could enhance FNO performance?

3. It would be beneficial to provide a clearer explanation and discussion of the theoretical results and their implications for readers without a background in mean-field theory.

**Limitations:**

Yes, the authors adequately addressed the limitations.

---

> ### Author Rebuttal · Authors · 2024-08-07
>
> Due to the character limit, we have addressed some questions in the global rebuttal. Please refer to our global rebuttal. We appreciate your attention and review.
>
> > Q1. The discovery in their work, ...
>
> The following part of the code is different from our He initialization, as it only initializes using the ‘nn.init.kaiming_uniform_’ function. The weights are initialized by a uniform distribution with different scales and the bias is not fixed to zero.
>
> ```python
> def reset_parameters(self):
>         if self.Hermitian:
>             for v in self.W.values():
>                 nn.init.kaiming_uniform_(v, a=math.sqrt(5))
>         else:
>             nn.init.kaiming_uniform_(self.W, a=math.sqrt(5))
>         if self.B is not None:
>             nn.init.kaiming_uniform_(self.B, a=math.sqrt(5))
> ```
> ---
>
> > Q2. If we treat FNO simply as a CNN ...
>
> Thank you very much for your insightful and valuable comments. We added this discussion of the similarities between CNN and FNO to the appendix. The following is our answer to this question and also the part of appendix.
>
> The iterated map for CNNs (Xiao et al., 2018) is obtained as follows:
> $$
> \Sigma^{(\ell)}\_{\alpha, \alpha'} = \frac{\sigma^2}{2r+1} \sum\_{\beta \in \mathrm{ker}} \mathbb{E}\_{\boldsymbol{H}\_{:. d} \sim \mathcal{N}(\boldsymbol{0}, \boldsymbol{\Sigma}^{(\ell-1)})}\left[ \phi(H\_{\alpha+\beta, d})\phi(H\_{\alpha'+\beta, d}) \right] + \sigma\_b^2
> $$
> where $2r + 1$ is the number of filter width and $\operatorname{ker} = \{ \beta \in \mathbb{Z} \mid |\beta| \leq r\}$ is the set of indices referring to the elements of the filter.
> When we consider the FNO without mode truncation ($K=N/2 + 1$), the propagation of the diagonal components $\Sigma\_{\alpha, \alpha}$ for any $\alpha \in [N]$ is equivalent to the propagation in a CNN with filter size $N$ performing global convolution.
>
> $$
> \begin{aligned}
>  \Sigma^{(\ell)}\_{\alpha, \alpha} &= \sigma^2 \sum\_{k=0}^{\frac{N}{2}} c\_k  \mathbb{E}\_{\boldsymbol{H}\_{:. d} \sim \mathcal{N}(\boldsymbol{0}, \boldsymbol{\Sigma}^{(\ell-1)})}\left[\left|\left[\mathcal{F}\phi\left(\boldsymbol{H}\_{:, d}\right)\right]\_{k}\right|^2 \right] \underbrace{\cos\left( \theta^{(k)}\_{\alpha, \alpha}\right)}\_{=1}  + \sigma^2\_b \\\\
>     &= \frac{\sigma^2}{N} \sum\_{\beta=0}^{N-1} \mathbb{E}\_{\boldsymbol{H}\_{:. d} \sim \mathcal{N}(\boldsymbol{0}, \boldsymbol{\Sigma}^{(\ell-1)})}\left[ \left|\phi(H\_{\beta, d})\right|^2\right] + \sigma^2\_b \\\\
>     &= \frac{\sigma^2}{N} \sum\_{\beta=0}^{N-1} \mathbb{E}\_{\boldsymbol{H}\_{:. d} \sim \mathcal{N}(\boldsymbol{0}, \boldsymbol{\Sigma}^{(\ell-1)})}\left[ \phi(H\_{\alpha+\beta, d})\phi(H\_{\alpha+\beta, d})\right] + \sigma^2\_b.
> \end{aligned}
> $$
> The first equality follows from Perseval's equality and the second from the periodic boundary condition. In contrast, the propagation of the off-diagonal components differs between CNN and FNO, and the iterated map is different even in the presence of mode truncation. Similarly, the iterated map during the backpropagation for FNO can be recognized as the global CNN (see Eq. (2.16) in (Xiao et al., 2018)) with $v\_{\beta} = \frac{1}{N}$.
> $$
> \tilde{\Sigma}\_{\alpha, \alpha}^{(\ell)} \approx \frac{1}{N^2} \sum\_{\beta, \beta'=0}^{N-1} \tilde{\Sigma}\_{\beta, \beta'}^{(\ell+1)}  \chi\_{c^*} \underbrace{\left(\sum\_{k=0}^{N/2} c\_k \cos\left( \theta^{(k)}\_{\beta-\beta', \alpha-\alpha} \right) \right)}_{=N \delta\_{\beta, \beta'}} = \chi\_{c^*} \sum\_{\beta=0}^{N-1} \frac{1}{N} \tilde{\Sigma}\_{\beta, \beta}^{(\ell+1)}.
> $$
> This equivalence suggests that the edge of chaos initialization (e.g., He initialization) is also valid for the FNO since the problems of gradient vanishing and explosion are determined by the diagonal components of $\tilde{\boldsymbol{\Sigma}}$.
>
> (Xiao et al., 2018) Xiao, Lechao, et al."Dynamical isometry and a mean field theory of cnns: How to train 10,000-layer vanilla convolutional neural networks."
>
> ---
>
> > Q3.  In line 122, it states: ... & The weights in spectral convolution ...
>
> For ease of handling in proofs, we define the complex weights $R^{(\ell)} \in \mathbb{C}^{D \times D \times K}$ of the $\ell$-th layer of the FNO by splitting them into their real and complex parts. We further divide these weights along the mode dimension using the index k, as $R^{(\ell)} = \bigg|\bigg|\_{k=1}^K \Theta^{(\ell, k)} + \sqrt{-1} \Xi^{(\ell, k)}$ where $\bigg|\bigg|$ denotes the concatenation operation along the mode axis k.
> For all $\ell \in [L]$ and $k \in [K]$, every element of the weights $\Theta^{(\ell, k)}$ and $\Xi^{(\ell, k)}$ is independently initialized from \$\mathcal{N}(0, \sigma^2/2D)$.
>
> ---
>
> > Q4: The FNO is implemented with 𝐿=64 layers,  ... Is such a large number of layers practical in real-world applications?
>
> Networks with a large number of layers are practical for real-world applications. For instance, as demonstrated in Appendix D.2, deeper FNOs with 16 or 32 layers achieved better performance than the conventional 4-layer FNO in learning complex operators. The study by Tran et al. (2023) on F-FNO also explores the training of relatively deep FNOs. In the field of PDE solving, it is possible to generate a large amount of data through numerical computation. Therefore, we believe that in the future, architectures with larger model capacities will be employed for more complex operator approximations and the construction of large pre-trained models as in the fields of CV or NLP. Our theory provides valuable insights for such multi-layer FNOs and architectures that partially use FNO layers.
>
> ---
>
> > Q5. Some abbreviations ...
>
> Thank you for pointing this out. We have revised this point.
>
> ---
>
> > Q6. Given your insights into FNO expressivity and trainability, do you have any thoughts on potential architectural improvements or variations that could enhance FNO performance?
>
> It seems difficult to obtain suggestions for new architectures. We may be able to give theoretical guarantees for newly proposed architectures.

---

> > ### Comment · Reviewer_gSyo · 2024-08-09
> >
> > Thank you for the reply. Some of my concerns have been addressed. However, I still feel that the impact of this work might be limited, and the practical side of this work needs to be further explored. I have adjusted my score.

---

> > > ### Author Response · Authors · 2024-08-10
> > >
> > > Dear Reviewer gSyo,
> > >
> > > Thank you very much for your thoughtful feedback and for adjusting your score. We appreciate your recognition of the improvements we have made. Your insights are invaluable, and we are grateful for your guidance in enhancing the quality and relevance of our work.

---

### Official Review · Reviewer_ePA1 · 2024-07-09

**Soundness:** 2
**Presentation:** 2
**Contribution:** 2
**Rating:** 5
**Confidence:** 3

**Summary:**

This paper leverages mean-field theory, developed for neural networks, to study the expressivity and trainability of random Fourier neural operators (FNOs). It provides practical insights for the appropriate initialization of FNOs.

**Strengths:**

The authors address a significant question in the neural operators literature: why does the performance of Fourier neural operators degrade as the number of layers increases? The authors take a first step toward answering this question by adopting an original theoretical perspective.

**Weaknesses:**

- The main body of the paper is not self-contained. Additionally, the technical section can be difficult to follow due to the need for back-and-forth referencing between multiple sections and references. Overall, the paper could benefit greatly from improved clarity. To enhance readability, I recommend reorganizing some parts of the text.
  - For example, the covariance matrix $\Sigma_{\alpha,\alpha'}^{(l)}$ should be introduced in section 3.2 where it is first examined.
  - At first glance, one might question if the iterated map in equation (6) is a definition. However, its origin becomes clear upon reviewing the supplementary material.
  - Lemma A.1 should be included in the main text, as it is a central result.
  - Some parts, such as the detailed description of the datasets, can be moved to the appendix to make space.
- I recommend providing further motivation for all concepts and referencing previous works where appropriate. For instance, the importance of studying the fixed point of the iterated map should be emphasized. Most importantly, please refer to the supplementary material for proofs when available.
- The proposed study focuses solely on the expressivity of random FNO at initialization, a rather limited scenario. This limitation should be more explicitly emphasized, particularly in the title. I am not convinced that the findings offer practical insights into the stable training of FNO, despite the preliminary numerical experiments provided. Additionally, the term _trainability_, as used by the authors, only pertains to the vanishing or exploding gradients of random FNO.
- One might question the impact of the optimizer, the initial learning rate, and its scheduler on the results. An ablation study addressing these factors would have been beneficial.

**Questions:**

- Does the fixed point of the iterated map always exist?
- The representation of Fig.1(a) is unclear. Does it depict the gradient norm averaged over the training epochs? I assume it shows the values only at initialization, averaged over multiple seeds. Additionally, is the standard deviation relatively small? Clarification on these points would be helpful.

**Limitations:**

I understand that error bars are omitted from the figures for the sake of clarity. However, including standard deviations in the tables would enhance the presentation of the results. Additionally, it is concerning that the results are averaged over only three seeds, which is insufficient for robust statistical analysis.

---

> ### Author Rebuttal · Authors · 2024-08-07
>
> Due to questions similar to those from other reviewers, we have addressed these in the global rebuttal. Please refer to our global rebuttal. We appreciate your attention and review.
>
> > Q1. The covariance matrix should be introduced in section 3.2.
>
> The covariance $\Sigma_{\alpha, \alpha’}^{(\ell)}$ is defined in Section 3.1, lines 135-136. For further clarity, definitions have also been added to the statements in Lemma 3.1 (Section 3.2).
>
> ---
>
> > Q2. Lemma A.1 should be included in the main text. Some parts, such as detailed description of datasets can be moved to the appendix.
>
> I appreciate your valuable suggestions. We moved Lemma A.1 to the line 155 in the main and some details about the dataset to the appendix.
>
> ---
>
> > Q3. The proposed study focuses solely on the expressivity of random FNO at initialization, a rather limited scenario. This limitation should be more explicitly emphasized, particularly in the title.
>
> Thank you for your helpful suggestions. We are considering the following titles. Which proposed title adequately reflects the scenario limitation?
> * Understanding the expressivity and trainability of Fourier Neural Operator at initialization
> * Understanding the impact of initialization on expressivity and trainability in Fourier Neural Operator
> * Understanding the expressivity and trainability of Mean Field Fourier Neural Operator
>
> ---
>
> > Q4. I am not convinced that the findings offer practical insights into the stable training of FNO, despite the preliminary numerical experiments provided.
>
> Our theory provides a universal insight, independent of the dataset, loss function, optimizer, and scheduler. While it does not offer sufficient conditions for stable training, it does provide a necessary condition for achieving stability in training. We believe that our results are of practical importance as they prevent unnecessary exploration of initialization. We have revisited the following.
>
> **line 44:** This discovery provides the necessary conditions (not sufficient conditions) for stable training, which is practically significant as it eliminates the unnecessary initialization searches.
>
> **line 310:** While we do not provide sufficient conditions for stable training, we do offer one necessary condition for achieving stable training.
>
> ---
>
> > Q5. The term trainability, as used by the authors, only pertains to the vanishing or exploding gradients of random FNO.
>
> The term trainability is used in the following related references with the same meaning as ours:
> * Xiao, Lechao, et al. "Dynamical isometry and a mean field theory of CNNs: How to train 10,000-layer vanilla convolutional neural networks."
> * Yang, Ge, and Samuel Schoenholz. "Mean field residual networks: On the edge of chaos."
> * Yang, Greg, et al. "A Mean Field Theory of Batch Normalization."
> * Chen, Minmin, et al. "Dynamical isometry and a mean field theory of RNNs: Gating enables signal propagation in recurrent neural networks."
>
> In these references, **trainability** refers to a necessary condition for successful training that is determined by the architecture, independent of the dataset, loss function, optimizer, or scheduler, primarily focusing on discussions related to gradient vanishing/exploding. However, we acknowledge your point that, in general, the term **trainability** could be interpreted to encompass the overall optimization process and the success of training. Therefore, we have revised the following sentences in the introduction for clarity. Thank you for your objective and valuable feedback.
>
> **Line 29:** We analyze the exponential expressivity (how far representations of different positions can be pulled apart) and trainability (how much gradient vanishing/explosion on average) of the random FNO from the perspective of whether the network is ordered or chaotic.
>
> ---
>
> > Q6. One might question the impact of the optimizer, the initial learning rate, and its scheduler on the results. An ablation study addressing these factors would have been beneficial.
>
> Thank you for your valuable suggestions. We are currently conducting additional experiments. If the results are ready in time for the discussion period, we will report them in the thread. However, we would like to reiterate that our theory does not tackle the analysis of the optimization process and our results represent necessary conditions for stable training, independent of these factors. It is important to emphasize that the failure of learning due to inappropriate choices of optimizer, learning rate, or scheduler is not inconsistent with our findings.
>
> ---
>
> > Q7. Does the fixed point of the iterated map always exist?
>
> Lemma A.1 guarantees the existence of a fixed point for the FNO when the DCN has a fixed point with $(q^*, c^* = 1)$. The existence of the fixed point for the DCN depends on the activation function, so we need to check it on a case-by-case basis. For example, despite both being bounded activation functions, Tanh has a fixed point, whereas the sigmoid function does not. Among unbounded activation functions, ReLU has a fixed point, but softplus (a smoothed version of ReLU) does not. The fixed point analysis for bounded activation functions is conducted in [1], and the general case is discussed in [2].
>
> [1] Poole, Ben, et al. "Exponential expressivity in deep neural networks through transient chaos."
>
> [2] Roberts, Daniel A., Sho Yaida, and Boris Hanin. "The principles of deep learning theory. "
>
> ---
>
> > Q8. The representation of Fig.1(a) is unclear. ...
>
> Fig. 1(a) experimentally verified Theorem 3.4. Instead of averaging over training epochs, It averages over multiple seeds at initialization (not over training epochs), with a relatively small standard deviation. We revised the first sentence of the title of Fig. 1 as follows.
>
> **fig1- title:** Average gradient norm $\operatorname{Tr}(\tilde{\Sigma}^{(\ell)})$ at initialization over multiple seeds during the backpropagation of several FNOs plotted as a function of layer $\ell$

---

> > ### Comment · Reviewer_ePA1 · 2024-08-11
> >
> > I would like to thank the authors for their detailed answer. While most of my concerns have been addressed, I agree with Reviewer gSyo’s conclusions. Consequently, I have increased my score to 5.

---

> > > ### Author Response · Authors · 2024-08-13
> > > **Results of additional experiments**
> > >
> > > Below are the results of the experiment conducted with two different optimizers (SGD and RMSProp) on the Navier-Stokes dataset with $\nu=1 \times 10^{-5}$. The average results from three different seeds are presented in the table below. The learning rates for SGD and RMSProp were determined through grid search, with SGD set at 0.00001 and RMSProp at 0.0005.
> > > The table indicates that, similar to the Adam optimizer, the deep FNO can only be effectively trained around $\sigma^2 = 2.0$. This suggests that our theoretical results are consistent and independent of the choice of optimizer.
> > >
> > > | SGD (train loss) | 0.1    | 0.5    | 1.0   | 2.0   | 3.0   | 4.0   |
> > > |------------------|--------|--------|-------|-------|-------|-------|
> > > | 4                | 12.654 | 4.136  | 3.143 | 2.901 | 2.876 | 2.733 |
> > > | 8                | 21.237 | 12.560 | 4.143 | 2.493 | inf     | inf     |
> > > | 16               | 21.237 | 21.236 | 8.304 | 2.203 | inf     | inf     |
> > > | 32               | 21.237 | 21.237 | 8.191 | 1.957 | inf     | inf     |
> > >
> > > | RMSProp (train loss) | 0.1    | 0.5    | 1.0    | 2.0   | 3.0   | 4.0   |
> > > |----------------------|--------|--------|--------|-------|-------|-------|
> > > | 4                    | 0.205  | 0.191  | 0.179  | 0.176 | 0.168 | 0.174 |
> > > | 8                    | 0.214  | 0.225  | 0.216  | 0.173 | 0.185 | 0.197 |
> > > | 16                   | 21.237 | 21.237 | 0.210  | 0.172 | inf    | inf    |
> > > | 32                   | 21.237 | 21.237 | 21.237 | 0.196 | inf    | inf    |
> > >
> > >
> > > Dear Reviewer ePA1
> > >
> > > Thank you very much for your detailed feedback and for taking the time to consider our responses. We are pleased to hear that we were able to address most of your concerns and that you have decided to increase your score.
> > > We understand and respect your agreement with Reviewer gSyo’s conclusions. We believe that our work provides a solid foundation and has the potential to make meaningful contributions to the field. We are committed to further refining our research and exploring additional aspects to enhance its impact and practical applications.
> > > Your feedback has been invaluable in helping us improve our paper, and we are grateful for your constructive comments and support.

---

### Official Review · Reviewer_xhAf · 2024-07-10

**Soundness:** 3
**Presentation:** 3
**Contribution:** 4
**Rating:** 8
**Confidence:** 4

**Summary:**

This paper presents a comprehensive theoretical analysis of the Fourier Neural Operator (FNO) using mean-field theory. The main contributions to me at least are:

1. Establishing a mean-field theory for FNO, analyzing its behavior from an edge of chaos perspective.
2. Examining the ordered-chaos phase transition of random FNO based on weight distribution.
3. Identifying a connection between expressivity and trainability, showing that ordered and chaotic phases correspond to regions of vanishing and exploding gradients respectively.
4. Providing practical initialization requirements for stable FNO training.

**Strengths:**

1. Theoretical Depth: The paper provides a rigorous mathematical analysis of FNOs, which has been lacking in the literature. The authors adapt and extend existing mean-field theory techniques to the specific structure of FNOs.
2. Novel Insights: The paper reveals FNO-specific characteristics in the phase transition, induced by mode truncation, while also showing similarities with densely connected networks. This bridges the gap between FNOs and more traditional neural network architectures. This was very very insighting to me!
3. Practical Implications: The theoretical analysis leads to practical guidelines for initializing FNOs, addressing a known issue of training instability in deep FNOs. This demonstrates the real-world value of the theoretical work.
4. Comprehensive Experiments: The authors validate their theoretical findings through extensive experiments on various PDE-solving tasks, showing the practical relevance of their analysis.
5. Clarity: Despite the technical nature of the content, the paper is well-structured and clearly written, making it accessible to a broader audience in the machine learning community.

**Weaknesses:**

1. There are no specific weaknesses that I see however, while the experiments are comprehensive, the authors don't fully explore very deep FNOs or a wide range of hyperparameters (in particular sweeping over the number of modes). Additional experiments on larger-scale problems or more complex PDEs could further strengthen the empirical validation.
2. Insufficient analysis of the impact of different Fourier modes: The paper doesn't provide a detailed analysis of how the number and distribution of Fourier modes affect the expressivity and trainability. This is crucial as the choice of modes is a key hyperparameter in FNOs.
3. Insufficient exploration of error propagation: For time-dependent PDEs, there's limited analysis of how errors propagate through time steps and how this relates to the expressivity and trainability of FNOs.

**Questions:**

1. I know this paper tackled FNO's, but I am curious if this could easily be adapted to other variants of FNOs like U-FNO, F-FNO etc
2. The paper doesn't provide a detailed analysis of how different PDE characteristics (e.g., stiffness, non-linearity, multi-scale behavior) might affect the expressivity and trainability of FNOs. Could you explain more on this point?

**Limitations:**

The authors have described their limitations!

---

> ### Author Rebuttal · Authors · 2024-08-07
>
> > Q1. The authors don't fully explore very deep FNOs or a wide range of hyperparameters (in particular sweeping over the number of modes). Additional experiments on larger-scale problems or more complex PDEs could further strengthen the empirical validation.
>
> Thank you for your valuable suggestions. We are currently conducting additional experiments. If the results are ready in time for the discussion period, we will report them in the thread.
>
> ---
>
> > Q2. I know this paper tackled FNO's, but I am curious if this could easily be adapted to other variants of FNOs like U-FNO, F-FNO etc
>
> Our theory may be applicable to several variants of the FNO. Examples include the following:
> * geo-FNO: For geo-FNO, the influence of deformation is limited to the beginning and the end of the architecture, with the majority of the architecture consisting of the original FNO. Thus, our theory can be directly applied.
> * G-FNO: For G-FNO, similar results can be obtained if the initial distribution of the transformed weight $\hat{L}_{s_g} R$ by elements $s_g$ from the stabilizer $S_g$ satisfies the assumptions of our theory.
> * WNO: Since the wavelet transform is a linear operator, the same analysis can potentially be applied to the WNO.
>
> Conversely, applying our theory directly to the architectures of U-FNO and F-FNO is challenging due to the presence of skip-connections. In architectures with skip-connections, the degeneration of input information in the ordered phase does not occur, and chaotic behavior is expected to be prominent. Therefore, it is necessary to analyze the behavior of the input in the chaotic phase in detail, rather than focusing on the discussion of the fixed point. It may be possible to extend our theory by incorporating mean-field theory for DCNs with skip-connections (Yang & Schoenholz, 2017) alongside our theoretical framework. Once the discussion on skip-connections is resolved, similar arguments can be made as follows:
> * U-FNO: For U-FNO, it is considered possible to discuss by integrating the results related to CNN, DCN, and simplified FNO, as discussed in Section 3.3 on the original FNO. Note that skip-connections are included in U-Net.
> * F-FNO: The factorized Fourier convolution module (Eq. 8) can be discussed similarly to the conventional FNO. However, since the output of the inverse FFT is summed in the dimensional direction, it seems appropriate to consider the weight scale as 1/D.
>
> (Yang & Schoenholz, 2017) Yang, Ge, and Samuel Schoenholz. "Mean field residual networks: On the edge of chaos." Advances in neural information processing systems 30 (2017).
>
> ---
>
> > Q3. The paper doesn't provide a detailed analysis of how different PDE characteristics (e.g., stiffness, non-linearity, multi-scale behavior) might affect the expressivity and trainability of FNOs. Could you explain more on this point?
>
> In terms of expressivity, it depends on the architecture of the FNO and not on the characteristics of the dataset (PDE). Regarding trainability, our theory provides a universal insight, independent of the dataset, loss function, optimizer, and scheduler. While it does not offer sufficient conditions for stable training, it does provide a necessary condition for achieving stability in training. As discussed in Appendix D, the characteristics of the dataset may, for example, lead to strong non-convexity problems in the loss landscape, but this is not explained by our theory. Our findings suggest that the acceptable initial variance of the weights is already limited by the number of layers, so these problems need to be addressed by optimizers, schedulers, devising loss functions, introducing regularization, etc.

---

> > ### Comment · Reviewer_xhAf · 2024-08-09
> > **Response to the authors**
> >
> > Thanks for running some additional experiments for deeper FNOs and a wider range of hyperparameters. The potential applicability of your theory to various FNO variants (geo-FNO, G-FNO, WNO) is insightful, and your explanation of the challenges with U-FNO and F-FNO due to skip-connections is clear! Super cool work. The suggestion to incorporate mean-field theory for DCNs with skip-connections is a promising direction for future work. Overall, I thank the authors for clarifiing several key points and highlighted potential directions for extending this work, strengthening the paper's contributions to understanding FNOs! I will keep my score however: )but great work!

---

> > > ### Author Response · Authors · 2024-08-10
> > >
> > > Dear Reviewer xhAf,
> > >
> > > Thank you very much for your positive feedback. Your encouraging words and recognition of our efforts to clarify key points and highlight potential directions for extending this work mean a lot to us. We are grateful for your high score and your constructive feedback, which have significantly contributed to strengthening our paper.

---

### Official Review · Reviewer_Jbgf · 2024-07-12

**Soundness:** 3
**Presentation:** 3
**Contribution:** 3
**Rating:** 7
**Confidence:** 3

**Summary:**

This work explores the expressivity and trainability of Fourier Neural Operators (FNOs) by establishing a mean-field theory and analyzing their behavior from an edge-of-chaos perspective.

**Strengths:**

The study examines the ordered-chaos phase transition of the network based on weight distribution, highlighting unique characteristics induced by mode truncation and similarities to densely connected networks. It also identifies a connection between expressivity and trainability, noting that ordered and chaotic phases correspond to regions of vanishing and exploding gradients, respectively. This insight provides a practical prerequisite for the stable training of FNOs. The experimental results support the theoretical findings.

**Weaknesses:**

1. The summary of related work can be further improved, particularly by including recent neural operator architectures from recent conferences or journals.
2. While Figure 1.(c) confirms that the original setup for initialization is reasonable, our experience suggests that the performance of FNOs in both training and test accuracy is highly dependent on the initialization variances for different hyperparameters and tasks. The derivation and results in this paper do not provide sufficient insights into this issue.
3. The code is not available.

**Questions:**

1. Can the derivation in this paper for FNOs be generalized to more general neural operators?
2. Is it possible to identify (approximately) how the initialization depends on different datasets?

**Limitations:**

Yes.

---

> ### Author Rebuttal · Authors · 2024-08-07
>
> > Q1. The summary of related work can be improved, particularly by including recent neural operator architectures
>
> We have revised Section 2.1 as follows. (Only newly added references are cited in numbered form.)
>
> **line 48:** The FNO (Li et al., 2021) is one of the well-established methods for solving PDEs across many scientific problems (Hwang et al., 2022; Jiang et al., 2021; Pathak et al., 2022; [1, 2]).
>
> **lines 58 - 71:** Several FNO variants have been developed to address specific challenges, such as geo-FNO (Li et al., 2022a) for irregular regions and group equivariant FNO (G-FNO) (Helwig et al., 2023), which maintains equivariance to rotation, reflection, and translation. U-NO[3] and U-FNO[4] integrate FNO with U-Net for multiscale modeling. Additionally, WNO[5] utilizes wavelet bases, while CFNO[6] enhances the use of geometric relations between different fields and field components through Clifford algebras. Adaptive FNO[7, 8] and F-FNO (Tran et al., 2022) have improved computational and memory efficiency through incremental learning and architectural modifications. Other approaches for improving performance include methods with increasing physical inductive bias[9], data augmentation[10], and a variance-preserving weight initialization scheme (Poli et al., 2022). While numerous new models and learning methods have been proposed, relatively little research has been conducted to understand the intrinsic nature of these methods. Issues such as spectral bias[7] and training instability (Tran et al., 2022) have been reported.
>
> [1] Yang, Y., et al. Seismic wave propagation and inversion with neural operators.
>
> [2] Wen, G., et al. Accelerating carbon capture and storage modeling using fourier neural operators.
>
> [3] Rahman, Md Ashiqur, et al. U-no: U-shaped neural operators
>
> [4] Wen, Gege, et al. U-FNO—An enhanced Fourier neural operator-based deep-learning model for multiphase flow
>
> [5] Tripura, Tapas, et al. Wavelet neural operator: a neural operator for parametric partial differential equations
>
> [6] Brandstetter, Johannes, et al., Clifford Neural Layers for PDE Modeling
>
> [7] Zhao, Jiawei, et al. Incremental spectral learning in Fourier neural operator.
>
> [8] Guibas, John, et al. Adaptive fourier neural operators: Efficient token mixers for transformers.
>
> [9] Li, Zongyi, et al. Physics-informed neural operator for learning partial differential equations.
>
> [10] Brandstetter, Johannes, et al. Lie point symmetry data augmentation for neural PDE solvers.
>
> ---
>
> > Q2. The code is not available.
>
> The paper does not propose any new models or learning methods and most of the code used in the experiments is based on the code of PDEBench. The only modifications to the model are to multiply the outputs (variable ‘out_ft’ in code of class FNO1d and FNO2d) corresponding to mode k=0, N/2 by $\sqrt{2}$ and to initialize the weights by Gaussian distribution, as described in Section 3.1.
> We added the same information as above to the detailed experimental setup description in the paper.
>
> ---
>
> > Q3. Can the derivation in this paper for FNOs be generalized to more general neural operators?
>
> Our theory may be applicable to several variants of the FNO. Examples include the following:
> * geo-FNO: For geo-FNO, the influence of deformation is limited to the beginning and the end of the architecture, with the majority of the architecture consisting of the original FNO. Thus, our theory can be directly applied.
> * G-FNO: For G-FNO, similar results can be obtained if the initial distribution of the transformed weight $\hat{L}_{s_g} R$ by elements $s_g$ from the stabilizer $S_g$ satisfies the assumptions of our theory.
> * WNO: Since the wavelet transform is a linear operator, the same analysis can potentially be applied to the WNO.
>
> Conversely, applying our theory directly to the architectures of U-FNO and F-FNO is challenging due to the presence of skip-connections. In architectures with skip-connections, the degeneration of input information in the ordered phase does not occur, and chaotic behavior is expected to be prominent. Therefore, it is necessary to analyze the behavior of the input in the chaotic phase in detail, rather than focusing on the discussion of the fixed point. It may be possible to extend our theory by incorporating mean-field theory for DCNs with skip-connections (Yang & Schoenholz, 2017) alongside our theoretical framework. Once the discussion on skip-connections is resolved, similar arguments can be made as follows:
> * U-FNO: For U-FNO, it is considered possible to discuss by integrating the results related to CNN, DCN, and simplified FNO, as discussed in Section 3.3 on the original FNO. Note that skip-connections are included in U-Net.
> * F-FNO: The factorized Fourier convolution module (Eq. 8) can be discussed similarly to the conventional FNO. However, since the output of the inverse FFT is summed in the dimensional direction, it seems appropriate to consider the weight scale as 1/D.
>
> ---
>
> > Q4. Is it possible to identify (approximately) how the initialization depends on different datasets?
>
> Our theory and experiments provide insight that, regardless of the dataset, loss function, optimizer, or scheduler, initialization around $\sigma^2 = 2.0$ is one of the necessary conditions (not sufficient condition) for stable training, especially for deep FNOs.
>
> At the same time, we also agree with your experience that the performance is dependent on hyperparameters and datasets (tasks). As discussed in Appendix D, underfitting caused by a loss landscape with strong non-convexity and overfitting caused by an inappropriate setting of model capacity relative to the amount of data are both hyperparameter- and data-dependent. Our findings suggest that the acceptable initial variance of the weights is already limited by the number of layers, so these problems need to be addressed by optimisers, schedulers, devising loss functions, introducing regularization, etc. Thank you for your valuable comments.

---

> > ### Comment · Reviewer_Jbgf · 2024-08-11
> >
> > Thanks for those detailed responses for possible new references and two questions. Now, I have better understanding about the limit and potential of this theoretical analysis. I will keep my score.

---

> > > ### Author Response · Authors · 2024-08-13
> > >
> > > Dear Reviewer Jbgf
> > >
> > > Thank you very much for your thoughtful feedback and for taking the time to review our responses in detail. We are glad to hear that our explanations have helped clarify the limits and potential of our theoretical analysis. Your insights have been instrumental in refining our understanding and presentation of the research. We are committed to further exploring the potential applications and implications of our findings to advance the field.

---

### Author Rebuttal · Authors · 2024-08-07

> Q1. **(reviewer ePA1)** I recommend providing further motivation for all concepts and referencing previous works where appropriate.
**(reviewer gSyo)** It would be beneficial to provide a clearer explanation and discussion of the theoretical results and their implications for readers without a background in mean-field theory.

We have included additional texts on the motivation for fixed point analysis and references to proofs located in their respective appendices. Is this explanation sufficiently clear for our readers? If you have any further suggestions or areas where you believe improvements could be made, please do not hesitate to inform us. Your feedback is greatly appreciated.

**line 152:** By linearizing the dynamics of signal propagation around this fixed point and analyzing the stability and rate of convergence to the fixed point, we can determine the depth to which each component of the input can propagate.

**line 155:** To analyze the stability and convergence rate, we linearly approximate the C-map around the fixed point $\Sigma^*$, i.e., $\mathcal{C}(\Sigma) \approx \Sigma^* + J\_{\Sigma^*} (\Sigma - \Sigma^*)$, where $J\_{\Sigma^*}$ is the Jacobian linear map of the iterated map defined in Eq. (15).

**line 161:** From Lemma A.4, $K-1$ matrices in $\{ \psi^{(k)} \}\_{k \in [K] \backslash \{0\}}$ are eigenbases with the eigenvalue $\chi_{c^*}$ of the Jacobian linear map. From Lemma A.5, the matrix $\psi$ is the eigenbases with the eigenvalue $\chi$ of the Jacobian linear map.
Since the rank of the Jacobian linear map is at most K (Lemma A.3), the deviation from the fixed point $\Sigma^{(\ell)} - \Sigma^*$ is spanned by K-dimensional eigenspace $\operatorname{span}\left(\\{\psi^{(k)} \\}_{k \in [K] \backslash \{0\}} \cup \\{ \psi \\} \right)$.

We have also added a discussion on the similarity with CNN to improve the interpretability of the results of Theorem 3.3 and 3.4.

---

> Q2. **(reviewer gSyo)** The significance of the theoretical aspects that FNOs share similarities with CNNs should be further discussed.

We have revised the text to emphasize that FNO has similarities not only with DCN but also with CNN. In particular, we have revised the text in Section 3.2 as follows.

**line 154:** (Xiao et al., 2018) showed that any fixed point for the iterated map of the DCN is also a fixed point for that of CNN.

**line 155:** Lemma A.1 means that the fixed point for the iterated map $\Sigma^*$ of the DCN serves as a fixed point for the iterated map of the simplified FNO (as well as CNN).

**line 178:** The convergence rates $\chi_{q^*}$ and $\chi_{c^*}$ are the same as the convergence rates of the variance and correlation to the fixed point for DCN (Schoenholz et al., 2016) and CNN (Xiao et al., 2018).

**line 182:** CNN possess diagonal eigenspaces associated with eigenvalues $\chi_{q^*}$ and non-diagonal eigenspaces associated with eigenvalues $\chi_{c^*}$. In contrast, FNOs without mode truncation exhibit a similarity, possessing eigenspaces $\chi_{q^*}$ for zero-frequency and eigenspaces $\chi_{c^*}$ for k-frequencies with diagonal components removed.

**line 200:** Despite the architectural and iterative map differences between FNO, DCN, and CNN, Theorems 3.3 and 3.4 demonstrate the similarity in the random behavior of FNO, DCN, and CNN.

---

### Decision · Program_Chairs · 2024-09-25

**Decision:**

Accept (poster)

**Comment:**

The article investigates the expressivity and trainability of Fourier Neural Operator.

Reviewers find the article addresses a significant question and that it offers theoretical depth, novel insight, practical implications, comprehensive experiments, and clarity. In particular, that it offered rigorous analysis of FNO and insights into practical aspects for training. Criticism included that the discussion of related work and discussion of certain experimental setups could be improved, that parts of the work were not self contained, and that the code was not made available. Authors offered additions to the references, additional experiments, and also explained that the article does not propose any new models or learning methods and the code used is based on other existing code. The responses clarified many of the questions and motivated reviewers to maintain positive scores or increase them.

At the end of the discussion period all reviewers recommend (borderline, weak, strong) accept. In view of the positive assessment at conclusion of the discussion, I recommend accept.